# Convex optimization based on global lower second-order models

**Nikita Doikov***
Catholic University of Louvain,
Louvain-la-Neuve, Belgium
Nikita.Doikov@uclouvain.be

**Yurii Nesterov†**
Catholic University of Louvain,
Louvain-la-Neuve, Belgium
Yurii.Nesterov@uclouvain.be

## Abstract

In this paper, we present new second-order algorithms for composite convex optimization, called Contracting-domain Newton methods. These algorithms are affine-invariant and based on global second-order lower approximation for the smooth component of the objective. Our approach has an interpretation both as a second-order generalization of the conditional gradient method, or as a variant of trust-region scheme. Under the assumption, that the problem domain is bounded, we prove $\mathcal{O}(1/k^2)$ global rate of convergence in functional residual, where $k$ is the iteration counter, minimizing convex functions with Lipschitz continuous Hessian. This significantly improves the previously known bound $\mathcal{O}(1/k)$ for this type of algorithms. Additionally, we propose a stochastic extension of our method, and present computational results for solving empirical risk minimization problem.

## 1 Introduction

Classical Newton method is one of the most popular optimization schemes for solving ill-conditioned problems. The method has very fast quadratic convergence, provided that the starting point is sufficiently close to the optimum [3, 22, 31]. However, the questions related to its global behaviour for a wide class of functions are still open, being in the area of active research.

The significant progress in this direction was made after [33], where Cubic regularization of Newton method with its global complexity bounds were justified. The main idea of [33] is to use a global *upper* approximation model of the objective, which is the second-order Taylor's polynomial augmented by a cubic term. The next point in the iteration process is defined as the minimizer of this model. Cubic Newton attains global convergence for convex functions with Lipschitz continuous Hessian. The rate of convergence in functional residual is of the order $\mathcal{O}(1/k^2)$ (here and later on, $k$ is the iteration counter). This is much faster than the classical $\mathcal{O}(1/k)$ rate of the Gradient Method [31]. Later on, accelerated [27], adaptive [7, 8] and universal [17, 12, 18] second-order schemes based on cubic regularization were developed. Randomized versions of Cubic Newton, suitable for solving high-dimensional problems were proposed in [13, 19].

Another line of results on global convergence of Newton method is mainly related to the framework of self-concordant functions [32, 31]. This class is affine-invariant. From the global perspective, it provides us with an *upper* second-order approximation of the objective, which naturally leads to the Damped Newton Method. Several new results are related to its analysis for generalized self-concordant functions [2, 38], and the notion of Hessian stability [23]. However, for more refined problem classes, we can often obtain much better complexity estimates, by using the cubic regularization technique [14].

In this paper, we investigate a different approach, which is motivated by a new global second-order *lower* model of the objective function, introduced in Section 3.

We incorporate this model into a new second-order optimization algorithm, called Contracting-Domain Newton Method (Section 4). At every iteration, it minimizes a lower approximation of the smooth component of the objective, augmented by a composite term. The next point is defined as a convex combination of the minimizer, and the previous point. By its nature, it is similar to the scheme of Conditional Gradient Method (or, Frank-Wolfe algorithm, [15, 30]). Under assumption of boundedness of the problem domain, for convex functions with Hölder continuous Hessian of degree $\nu \in [0, 1]$, we establish its $\mathcal{O}(1/k^{1+\nu})$ global rate of convergence in functional residual. In the case $\nu = 1$, for the class of convex function with Lipschitz continuous Hessian, this gives $\mathcal{O}(1/k^2)$ rate of convergence. As compared with Cubic Newton, the new method is affine-invariant and universal, since it does not depend on the norms and parameters of the problem class. When the composite component is strongly convex (with respect to arbitrary norm), we show $\mathcal{O}(1/k^{2+2\nu})$ rate for a universal scheme. If the parameters of problem class are known, we can prove a global linear convergence. We also provide different trust-region interpretations for our algorithm.

In Section 5, we present aggregated models, which accumulate second-order information into quadratic Estimating Functions [31]. This leads to another optimization process, called Aggregating Newton Method, with the global convergence of the same order $\mathcal{O}(1/k^{1+\nu})$ as for general convex case. The latter method can be seen as a second-order counterpart of the dual averaging gradient schemes [28, 29].

In Section 6, we consider the problem of finite-sum minimization. We propose stochastic extensions of our method. During the iterations of the basic variant, we need to increase the batch size for randomized estimates of gradients and Hessians up to the order $\mathcal{O}(k^4)$ and $\mathcal{O}(k^2)$ respectively. Using the *variance reduction* technique for the gradients, we reduce the batch size up to the level $\mathcal{O}(k^2)$ for both estimates. At the same time, the global convergence rate of the resulting methods is of the order $\mathcal{O}(1/k^2)$, as for general convex functions with Lipschitz continuous Hessian.

Section 7 contains numerical experiments. Section 8 contains some final remarks. All necessary proofs are provided in the supplementary material.

## 2 Problem formulation and notations

Our goal is to solve the following *composite* convex minimization problem:
$$\min_x F(x) \quad := \quad f(x) + \psi(x), \tag{1}$$
where $\psi : \mathbb{R}^n \to \mathbb{R} \cup \{+\infty\}$ is a *simple* proper closed convex function, and function $f$ is convex and twice continuously differentiable at every point $x \in \operatorname{dom} \psi$. Let us fix an arbitrary (possibly non-Euclidean) norm $\| \cdot \|$ on $\mathbb{R}^n$. We denote by $D$ the corresponding diameter of $\operatorname{dom} \psi$:
$$D \quad := \quad \sup_{x, y \in \operatorname{dom} \psi} \|x - y\|. \tag{2}$$
Our main assumption on problem (1) is that $\operatorname{dom} \psi$ is bounded:
$$D \quad < \quad +\infty. \tag{3}$$
The most important example of $\psi$ is $\{0, +\infty\}$-indicator of a simple compact convex set $Q = \operatorname{dom} \psi$. In particular, for a ball in $\| \cdot \|_p$-norm with $p \geq 1$, this is
$$\psi(x) \quad = \quad \begin{cases} 0, & \|x\|_p := \left( \sum_{i=1}^n |x^{(i)}|^p \right)^{1/p} \leq \frac{D}{2}, \\ +\infty, & \text{else.} \end{cases} \tag{4}$$
From the machine learning perspective, $D$ is usually considered as a *regularization parameter* in this setting. We denote by $\langle \cdot, \cdot \rangle$ the standard scalar product of two vectors, $x, y \in \mathbb{R}^n$:
$$\langle x, y \rangle \quad := \quad \sum_{i=1}^n x^{(i)} y^{(i)}.$$
For function $f$, we denote its gradient by $\nabla f(x) \in \mathbb{R}^n$, and its Hessian matrix by $\nabla^2 f(x) \in \mathbb{R}^{n \times n}$. Having fixed the norm $\| \cdot \|$ for primal variables $x \in \mathbb{R}^n$, the *dual* norm can be defined in the standard way:
$$\|s\|_* \quad := \quad \sup_{h \in \mathbb{R}^n : \|h\| \leq 1} \langle s, h \rangle.$$

The dual norm is necessary for measuring the size of gradients. For a matrix $A \in \mathbb{R}^{n \times n}$, we use the corresponding induced operator norm, defined as

$$\|A\| \quad := \quad \sup_{h \in \mathbb{R}^n : \|h\| \leq 1} \|Ah\|_*.$$

## 3  Second-order lower model of objective function

To characterize the complexity of problem (1), we need to introduce some assumptions on the growth of derivatives. Let us assume that the Hessian of $f$ is Hölder continuous of degree $\nu \in [0,1]$ on $\operatorname{dom} \psi$:

$$\|\nabla^2 f(x) - \nabla^2 f(y)\| \quad \leq \quad H_\nu \|x - y\|^\nu, \qquad x, y \in \operatorname{dom} \psi. \tag{5}$$

The actual parameters of this problem class may be unknown. However, we assume that for *some* $\nu \in [0,1]$ inequality (5) is satisfied with corresponding constant $0 \leq H_\nu < +\infty$. The direct consequence of (5) is the following global bounds for Taylor's approximation, for all $x, y \in \operatorname{dom} \psi$

$$\|\nabla f(y) - \nabla f(x) - \nabla^2 f(x)(y - x)\|_* \quad \leq \quad \frac{H_\nu \|y - x\|^{1+\nu}}{1+\nu}, \tag{6}$$

$$|f(y) - f(x) - \langle \nabla f(x), y - x \rangle - \tfrac{1}{2}\langle \nabla^2 f(x)(y - x), y - x \rangle| \quad \leq \quad \frac{H_\nu \|y - x\|^{2+\nu}}{(1+\nu)(2+\nu)}. \tag{7}$$

Recall, that in addition to (5), we assume that $f$ is *convex*:

$$f(y) \quad \geq \quad f(x) + \langle \nabla f(x), y - x \rangle, \qquad x, y \in \operatorname{dom} \psi. \tag{8}$$

Employing both smoothness and convexity, we are able to enhance this global lower bound, as follows.

---

**Lemma 1** *For all $x, y \in \operatorname{dom} \psi$ and $t \in [0,1]$, it holds*

$$f(y) \quad \geq \quad f(x) + \langle \nabla f(x), y - x \rangle + \tfrac{t}{2}\langle \nabla^2 f(x)(y - x), y - x \rangle - \frac{t^{1+\nu} H_\nu \|y - x\|^{2+\nu}}{(1+\nu)(2+\nu)}. \tag{9}$$

---

Note that the right-hand side of (9) is concave in $t \in [0,1]$, and for $t = 0$ we obtain the standard first-order lower bound. The maximization of (9) over $t$ gives

$$f(y) \quad \geq \quad f(x) + \langle \nabla f(x), y - x \rangle + \tfrac{\bar{\gamma}_{x,y}}{2}\langle \nabla^2 f(x)(y - x), y - x \rangle, \tag{10}$$

with

$$\bar{\gamma}_{x,y} \quad := \quad \frac{\nu}{1+\nu} \min\left\{1, \frac{(2+\nu)\langle \nabla^2 f(x)(y-x), y-x \rangle}{2H_\nu \|y-x\|^{2+\nu}}\right\}^{\frac{1}{\nu}}, \qquad x \neq y, \quad \nu \in (0,1].$$

Thus, (10) is always *tighter* than (8), employing additional *global second-order information*. The relationship between them is shown on Figure 1. Hence, it seems natural to incorporate the second-order lower bounds into optimization schemes.

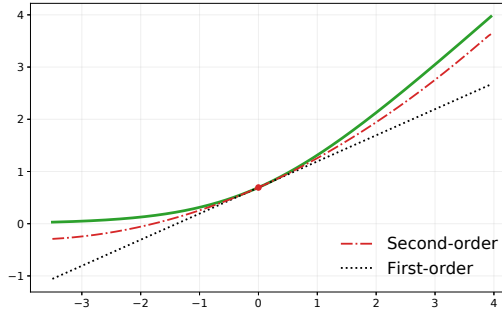

**Figure 1:** Global lower bounds for logistic regression loss, $f(x) = \log(1 + \exp(x))$.

# 4 Contracting-Domain Newton Methods

Let us introduce a general scheme of *Contracting-Domain Newton Method*, which is based on global second-order lower bounds. Note, that the right hand side of (10) is nonconvex in $y$. Hence, it can hardly be used directly in a computational algorithm. To tackle this issue, we use a sequence of contracting coefficients $\{\gamma_k\}_{k \geq 0}$. Each coefficient $\gamma_k \in (0, 1]$ can be seen as an appropriate substitute of $\bar{\gamma}_{x,y}$ in (10). Then, we minimize the corresponding global lower bound augmented by the composite component $\psi(\cdot)$. The next point is taken as a convex combination of the minimizer and the current point. Let us present this method formally, as Algorithm 1.

---
**Algorithm 1:** Contracting-Domain Newton Method, I

---
**Initialization.** Choose $x_0 \in \operatorname{dom} \psi$.
**Iteration** $k \geq 0$.
1: Pick up $\gamma_k \in (0, 1]$.
2: Compute
$$v_{k+1} \in \operatorname*{Argmin}_y \Big\{ \langle \nabla f(x_k), y - x_k \rangle + \tfrac{\gamma_k}{2} \langle \nabla^2 f(x_k)(y - x_k), y - x_k \rangle + \psi(y) \Big\}.$$
3: Set $x_{k+1} := x_k + \gamma_k(v_{k+1} - x_k)$.

---

There is a clear connection of this method with Frank-Wolfe algorithm, [15]. Indeed, instead of the standard first-order approximation (8), we use the lower global quadratic model. Thus, as compared with the gradient methods, every iteration of Algorithm 1 is more expensive. However, this is a standard situation with the second-order schemes (see the below discussion on the iteration complexity). At the same time, our method is *affine-invariant*, since it does not depend on the norms.

It is clear, that for $\gamma_k \equiv 1$ we obtain iterations of the classical Newton method. Its local quadratic convergence for composite optimization problems was established in [26]. However, for the global convergence, we need to adjust the contracting coefficients accordingly. To state the global convergence result, let us introduce the following linear *Estimating Functions* (see [31]):

$$\phi_k(x) \stackrel{\text{def}}{=} \sum_{i=1}^k a_i \big[ f(x_i) + \langle \nabla f(x_i), x - x_i \rangle + \psi(x) \big], \qquad \phi_k^* := \min_x \phi_k(x), \quad (11)$$

for the sequence of test points $\{x_k : x_k \in \operatorname{dom} \psi\}_{k \geq 1}$ and positive scaling coefficients $\{a_k\}_{k \geq 1}$. We relate them with contracting coefficients, as follows

$$\gamma_k := \frac{a_{k+1}}{A_{k+1}}, \qquad A_k \stackrel{\text{def}}{=} \sum_{i=1}^k a_i. \quad (12)$$

---
**Theorem 1** *Let $A_k := k^3$, and consequently, $\gamma_k := 1 - \left(\frac{k}{k+1}\right)^3 = \mathcal{O}\left(\frac{1}{k}\right)$. Then for the sequence $\{x_k\}_{k \geq 1}$ generated by Algorithm 1, we have*

$$F(x_k) - F^* \leq \ell_k \stackrel{\text{def}}{=} F(x_k) - \frac{\phi_k^*}{A_k} \leq \mathcal{O}\left(\frac{H_\nu D^{2+\nu}}{k^{1+\nu}}\right). \quad (13)$$

---

For the case $\nu = 1$ (convex functions with Lipschitz continuous Hessian), estimate (13) gives the convergence rate of the order $\mathcal{O}\left(\frac{1}{k^2}\right)$. This is the same rate, as we can achieve on this functional class by Cubic Regularization of Newton Method [33]. In accordance to (13), in order to obtain $\varepsilon$-accuracy in functional residual, $F(x_K) - F^* \leq \varepsilon$, it is enough to perform

$$K = \mathcal{O}\left(\left(\frac{H_\nu D^{2+\nu}}{\varepsilon}\right)^{1/(1+\nu)}\right) \quad (14)$$

iterations of Algorithm 1. In [17], there were proposed first *universal* second-order methods (which do not depend on parameters $\nu$ and $H_\nu$ of the problem class), having complexity guarantees of the same order (14). These methods are based on Cubic regularization and an adaptive search for estimating the regularization parameter at every iteration. It is important that Algorithm 1 is both universal and affine-invariant. Additionally, convergence result (13) provides us with a sequence $\{\ell_k\}_{k \geq 1}$ of computable *accuracy certificates*, which can be used as a stopping criterion of the method.

Now, let us assume that the composite component is *strongly convex* with parameter $\mu > 0$. Thus, for all $x, y \in \operatorname{dom} \psi$ and $\psi'(x) \in \partial \psi(x)$, it holds

$$\psi(y) \geq \psi(x) + \langle \psi'(x), y - x \rangle + \tfrac{\mu}{2} \|y - x\|^2. \quad (15)$$

In this situation, we are able to improve convergence estimate (13), as follows.

**Theorem 2** *Let $A_k := k^5$, and consequently, $\gamma_k := 1 - \left(\frac{k}{k+1}\right)^5 = \mathcal{O}\left(\frac{1}{k}\right)$. Then for the sequence $\{x_k\}_{k \geq 1}$ generated by Algorithm 1, we have*

$$F(x_k) - F^* \leq \ell_k \leq \mathcal{O}\left(\frac{H_\nu D^\nu}{\mu} \cdot \frac{H_\nu D^{2+\nu}}{k^{2+2\nu}}\right). \tag{16}$$

*Moreover, if the second-order condition number*

$$\omega_\nu \stackrel{\text{def}}{=} \left[\frac{H_\nu D^\nu}{(1+\nu)\mu}\right]^{\frac{1}{1+\nu}} \tag{17}$$

*is known, then, defining $A_k := (1 + \omega_\nu^{-1})^k$, $k \geq 1$, $A_0 := 0$, and $\gamma_k := \frac{1}{1+\omega_\nu}$, $k \geq 1$, $\gamma_0 := 1$, we obtain the global linear rate of convergence*

$$F(x_k) - F^* \leq \ell_k \leq \exp\left(-\frac{k-1}{1+\omega_\nu}\right) \cdot \frac{H_\nu D^{2+\nu}}{1+\nu}. \tag{18}$$

According to the estimate (18), in order to get $\varepsilon$-accuracy in function value, it is enough to perform

$$K = \mathcal{O}\left((1 + \omega_\nu) \cdot \log \frac{F(x_0) - F^*}{\varepsilon}\right)$$

iterations of the method. Hence, condition number $\omega_\nu$ plays the role of the main complexity factor. This rate corresponds to that one of Cubically Regularized Newton Method (see [11, 12]). At the same time, there exists a second variant of Contracting-Domain Newton Method, where the next point is defined by minimization of the full second-order model for the smooth component augmented by the composite term over the *contracted domain* (this explains the names of our methods).

---

**Algorithm 2:** Contracting-Domain Newton Method, II

---

**Initialization.** Choose $x_0 \in \operatorname{dom}\psi$.
**Iteration** $k \geq 0$.
1: Pick up $\gamma_k \in (0, 1]$.
2: Denote
$$S_k(y) := \begin{cases} \psi(y), & y \in \gamma_k \operatorname{dom}\psi + (1 - \gamma_k)x_k, \\ +\infty, & \text{else.} \end{cases}$$
3: Compute
$$x_{k+1} \in \operatorname{Argmin}_y\left\{\langle \nabla f(x_k), y - x_k\rangle + \frac{1}{2}\langle \nabla^2 f(x_k)(y - x_k), y - x_k\rangle + S_k(y)\right\}.$$

---

Note, that Algorithm 1 admits similar representation as well. [3] Both methods produce the same sequences of points when $\psi(\cdot)$ is $\{0, +\infty\}$-indicator of a convex set. Otherwise, they are different. Using the same contraction technique, it was shown in [30] that the classical Frank-Wolfe algorithm can be extended onto the case of the composite optimization problems. Additionally, the second-order *Contracting Trust-Region method* was proposed, which has the same form as Algorithm 2. However, its convergence rate was established only at the level $\mathcal{O}(\frac{1}{k})$. Here, we improve its rate as follows.

---

**Theorem 3** *Let $A_k := k^3$ and $\gamma_k := 1 - \left(\frac{k}{k+1}\right)^3 = \mathcal{O}\left(\frac{1}{k}\right)$. Then for the sequence $\{x_k\}_{k \geq 1}$ generated by Algorithm 2, we have*

$$F(x_k) - F^* \leq \ell_k \leq \mathcal{O}\left(\frac{H_\nu D^{2+\nu}}{k^{1+\nu}}\right). \tag{19}$$

---

This result is very similar to Theorem 1. However, the first algorithm can be accelerated on the class of strongly convex functions (see Theorem 2). Thus, it seems that it is more preferable.

Finally, let us consider an example, when the composite component $\psi(\cdot)$ is an $\ell_p$-ball, as in (4). Then, iterations of the method can be represented as

$$x_{k+1} \in x_k + \operatorname{Argmin}_h\left\{\langle \nabla f(x_k), h\rangle + \frac{1}{2}\langle \nabla^2 f(x_k)h, h\rangle : \left\|x_k + \frac{1}{\gamma_k}h\right\|_p \leq \frac{D}{2}\right\}. \tag{20}$$

In this form, it looks as a variant of Trust-Region scheme. To solve the subproblem in (20), we can use Interior Point Methods (e.g. Chapter 5 in [31]). See also [9], for techniques, developed for Trust-Region schemes. Usually, complexity of this step can be estimated as $\mathcal{O}(n^3)$ arithmetic operations,

which comes from the cost of computing a suitable factorization for the Hessian matrix. Alternatively, Hessian-free gradient methods can be applied, for computing an inexact step (see [6, 5]).

# 5    Aggregated second-order models

In this section, we propose more advanced second-order models, based on global lower bound (9). Using the same notation as before, consider a sequence of test points $\{x_k : x_k \in \operatorname{dom} \psi\}_{k \geq 0}$ and sequences of coefficients $\{a_k\}_{k \geq 1}, \{\gamma_k\}_{k \geq 0}$, satisfying the relations (12). Then, we can introduce the following *Quadratic Estimating Functions* (compare with definition (11)):

$$Q_k(x) \;\; \overset{\text{def}}{=} \;\; \sum_{i=0}^{k-1} a_{i+1}\Big[f(x_i) + \langle \nabla f(x_i), x - x_i \rangle + \tfrac{\gamma_i}{2}\langle \nabla^2 f(x_i)(x - x_i), x - x_i \rangle + \psi(x)\Big].$$

By (9), we have the main property of Estimating Functions being satisfied. Namely, for all $x \in \operatorname{dom} \psi$

$$A_k F(x) \;\; \overset{(9)}{\geq} \;\; Q_k(x) - \sum_{i=0}^{k-1} \tfrac{a_{i+1}\gamma_i^{1+\nu} H_\nu \|x - x_i\|^{2+\nu}}{(1+\nu)(2+\nu)}$$

$$\overset{(2)}{\geq} \;\; Q_k(x) - \tfrac{H_\nu D^{2+\nu}}{(1+\nu)(2+\nu)} \sum_{i=0}^{k-1} a_{i+1}\gamma_i^{1+\nu} \;\; =: \;\; Q_k(x) - \tfrac{C_k}{2}. \tag{21}$$

Therefore, if we would be able to guarantee for our test points the relation

$$Q_k^* \;\; := \;\; \min_x Q_k(x) \;\; \geq \;\; A_k F(x_k) - \tfrac{C_k}{2}, \tag{22}$$

then we could immediately obtain the global convergence in function value. Fortunately, relation (22) can be achieved by simple iterations.

---

**Algorithm 3:** Aggregating Newton Method

---

**Initialization.** Choose $x_0 \in \operatorname{dom} \psi$. Set $A_0 := 0$, $Q_0(x) \equiv 0$.
**Iteration** $k \geq 0$.
  1: Pick up $a_{k+1} > 0$. Set $A_{k+1} := A_k + a_{k+1}$ and $\gamma_k := \tfrac{a_{k+1}}{A_{k+1}}$.
  2: Update Estimating Function
     $Q_{k+1}(x) \;\equiv\; Q_k(x) + a_{k+1}\big[f(x_k) + \langle \nabla f(x_k), x - x_k \rangle + \tfrac{\gamma_k}{2}\langle \nabla^2 f(x_k)(x - x_k), x - x_k \rangle + \psi(x)\big].$
  3: Compute
$$v_{k+1} \;\; \in \;\; \operatorname*{Argmin}_x Q_{k+1}(x).$$
  4: Set $x_{k+1} := x_k + \gamma_k(v_{k+1} - x_k)$.

---

Clearly, the most complicated part of this process is Step 3, which is computation of the minimum of Estimating Function. However, the complexity of this step remains the same, as that one for Contracting-Domain Newton Method. We obtain the following convergence result.

---

**Theorem 4** *For the sequence $\{x_k\}_{k \geq 1}$ generated by Algorithm 3, relation* (22) *is satisfied. Consequently, for the choice $A_k := k^3$, we obtain*

$$F(x_k) - F^* \;\; \overset{(21)}{\leq} \;\; F(x_k) - \tfrac{Q_k^*}{A_k} + \tfrac{C_k}{2A_k} \;\; \overset{(22)}{\leq} \;\; \tfrac{C_k}{A_k} \;\; \leq \;\; \mathcal{O}\big(\tfrac{H_\nu D^{2+\nu}}{k^{1+\nu}}\big). \tag{23}$$

---

Now, for the accuracy certificate we have new expression $\bar{\ell}_k := F(x_k) - \tfrac{Q_k^*}{A_k} + \tfrac{C_k}{2A_k}$. The value of $Q_k^*$ is available within the method directly. However, in order to compute $\bar{\ell}_k$ in practice, some estimate for $C_k$ is required. Note, that for the given choice of coefficients $A_k := k^3$, we have $a_k = \mathcal{O}(k^2)$ and $\gamma_k = \mathcal{O}(\tfrac{1}{k})$. Therefore, new information enters into the model with increasing weights, which seems to be natural.

# 6    Stochastic finite-sum minimization

In this section, we consider the case when the smooth part $f$ of the objective (1) is represented as a sum of $M$ convex twice-differentiable components,

$$f(x) \;\; := \;\; \tfrac{1}{M} \sum_{i=1}^{M} f_i(x). \tag{24}$$

This setting appears in many machine learning applications, such as *empirical risk minimization*. Often, the number $M$ is very big. Thus, it becomes expensive to evaluate the whole gradient or the Hessian at every iteration. Hence, *stochastic* or *incremental* methods are the methods of choice in this situation. See [4] for a survey of first-order incremental methods. The Newton-type Incremental Method with superlinear local convergence was proposed in [35]. Local linear rate of stochastic Newton methods was studied in [25]. Global convergence of sub-sampled Newton schemes, based on Damped iterations, and on Cubic regularization, was established in [36, 24, 39].

The basic idea of stochastic algorithms is to substitute the true gradients and Hessians by some random unbiased estimators $g_k$, and $H_k$, respectively, with $\mathbb{E}[g_k] = \nabla f(x_k)$ and $\mathbb{E}[H_k] = \nabla^2 f(x_k)$.

First, let us consider the simplest estimation strategy. At iteration $k$, we sample uniformly and independently two subsets of indices $S_k^g, S_k^H \subseteq \{1, \ldots, M\}$. Their sizes are $m_k^g := |S_k^g|$ and $m_k^H := |S_k^H|$, which are possibly different. Then, in Algorithm 1, we can use the following random estimators:

$$g_k \quad := \quad \frac{1}{m_k^g} \sum_{i \in S_k^g} \nabla f_i(x_k), \qquad H_k \quad := \quad \frac{1}{m_k^H} \sum_{i \in S_k^H} \nabla^2 f_i(x_k). \qquad (25)$$

Let us present for this process a result on its global convergence. Note that in this section, we use the standard Euclidean norm for vectors and the corresponding induced spectral norm for matrices.

---

**Theorem 5** *Let each component $f_i(\cdot)$ be Lipschitz continuous on $\operatorname{dom}\psi$ with constant $L_0$, and have Lipschitz continuous gradients and Hessians on $\operatorname{dom}\psi$ with constants $L_1$ and $L_2$, respectively. Let $\gamma_k := 1 - \left(\frac{k}{k+1}\right)^3 = \mathcal{O}\left(\frac{1}{k}\right)$. Set*

$$m_k^g \quad := \quad 1/\gamma_k^4, \qquad m_k^H \quad := \quad 1/\gamma_k^2. \qquad (26)$$

*Then, for the iterations $\{x_k\}_{k \geq 1}$ of Algorithm (1), based on estimators (25), it holds*

$$\mathbb{E}[F(x_k) - F^*] \quad \leq \quad \mathcal{O}\left(\frac{L_2 D^3 + L_1 D^2(1 + \log(n)) + L_0 D}{k^2}\right). \qquad (27)$$

---

Therefore, in order to solve our problem with $\varepsilon$-accuracy in expectation, $\mathbb{E}[F(x_K) - F^*] \leq \varepsilon$, we need to perform $K = \mathcal{O}\left(\frac{1}{\varepsilon^{1/2}}\right)$ iterations of the method. In this case, the total number of gradient and Hessian samples are $\mathcal{O}\left(\frac{1}{\varepsilon^{5/2}}\right)$ and $\mathcal{O}\left(\frac{1}{\varepsilon^{3/2}}\right)$, respectively. It is interesting that we need higher accuracy for estimating the gradients, which results in a bigger batch size.

To improve this result, we incorporate a simple *variance reduction* strategy for the gradients. This is a popular technique in stochastic convex optimization (see [37, 21, 10, 20, 1, 34, 16] and references therein). At some iterations, we recompute the full gradient. However, during the whole optimization process this happens logarithmic number of times in total. Let us denote by $\pi(k)$ the maximal *power of two*, which is less than or equal to $k$: $\pi(k) := 2^{\lfloor \log_2 k \rfloor}$, for $k > 0$, and define $\pi(0) := 0$. The entire scheme looks as follows.

---

**Algorithm 4:** Stochastic Variance-Reduced Contracting-Domain Newton

---

**Initialization.** Choose $x_0 \in \operatorname{dom}\psi$.
**Iteration** $k \geq 0$.
 1: Set anchor point $z_k := x_{\pi(k)}$.
 2: Sample random batch $S_k \subseteq \{1, \ldots, M\}$ of size $m_k$.
 3: Compute variance-reduced stochastic gradient
$$g_k \quad := \quad \frac{1}{m_k} \sum_{i \in S_k} \left(\nabla f_i(x_k) - \nabla f_i(z_k) + \nabla f(z_k)\right).$$
 4: Compute stochastic Hessian
$$H_k \quad := \quad \frac{1}{m_k} \sum_{i \in S_k} \nabla^2 f_i(x_k).$$
 5: Pick up $\gamma_k \in (0, 1]$.
 6: Perform the main step
$$x_{k+1} \quad \in \quad \operatorname*{Argmin}_{y} \left\{ \langle g_k, y - x_k \rangle + \tfrac{1}{2}\langle H_k(y - x_k), y - x_k \rangle + \gamma_k \psi\left(x_k + \tfrac{1}{\gamma_k}(y - x_k)\right) \right\}.$$

---

Note that this is just Algorithm 1 with random estimators $g_k$ and $H_k$ instead ot the true gradient and Hessian. The following global convergence result holds.

**Theorem 6** *Let each component $f_i(\cdot)$ have Lipschitz continuous gradients and Hessians on* $\operatorname{dom}\psi$ *with constants $L_1$ and $L_2$, respectively. Let $\gamma_k := 1 - \left(\frac{k}{k+1}\right)^3 = \mathcal{O}(\frac{1}{k})$. Set batch size*

$$m_k \quad := \quad 1/\gamma_k^2. \tag{28}$$

*Then, for all iterations $\{x_k\}_{k \geq 1}$ of Algorithm 4, we have*

$$\mathbb{E}[F(x_k) - F^*] \quad \leq \quad \mathcal{O}\left(\frac{L_2 D^3 + L_1 D^2(1 + \log(n)) + L_1^{1/2} D(F(x_0) - F^*)}{k^2}\right). \tag{29}$$

It is thanks to the variance reduction that we can use the same batch size for both estimators now. To solve the problem with $\varepsilon$-accuracy in expectation, we need $K = \mathcal{O}\left(\frac{1}{\varepsilon^{1/2}}\right)$ iterations of the method. And the total number of gradient and Hessian samples during these iterations is $\mathcal{O}\left(\frac{1}{\varepsilon^{3/2}}\right)$.

## 7 Experiments

Let us demonstrate computational results for the problem of training Logistic Regression model, regularized by $\ell_2$-ball constraints. Thus, the smooth part of the objective has the finite-sum representation (24), each component is $f_i(x) := \log(1 + \exp(\langle a_i, x \rangle))$. The composite part is given by (4), with $p = 2$. Diameter $D$ plays the role of regularization parameter, while vectors $\{a_i : a_i \in \mathbb{R}^n\}_{i=1}^M$ are determined by the dataset[4]. First, we compare the performance of Contracting-Domain Newton Method (Algorithm 1) and Aggregating Newton Method (Algorithm 3) with first-order optimization schemes: Frank-Wolfe algorithm [15], the classical Gradient Method, and the Fast Gradient Method [29]. For the latter two we use a line-search at each iteration, to estimate the Lipschitz constant. The results are shown on Figure 2.

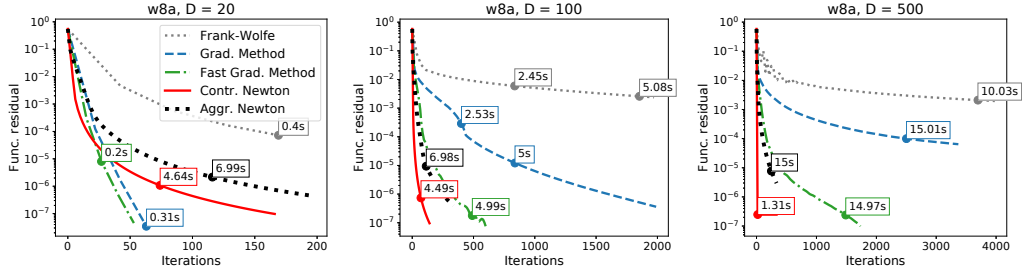

**Figure 2:** Training logistic regression, *w8a* $(M = 49749, n = 300)$.

We see, that for bigger $D$, it becomes harder to solve the optimization problem. Second-order methods demonstrate good performance both in terms of the iterations, and the total computational time. [5]

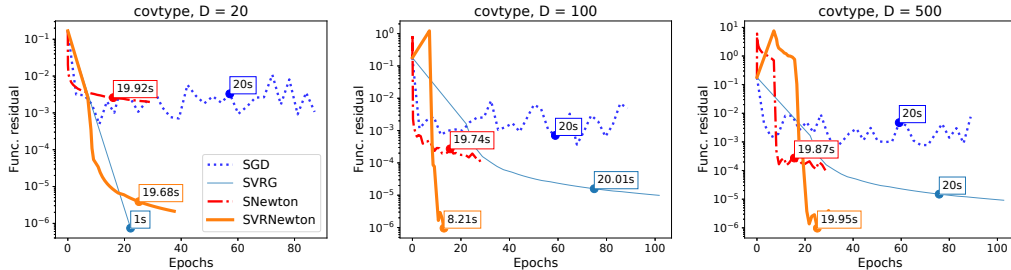

**Figure 3:** Stochastic methods for training logistic regression, *covtype* $(M = 581012, n = 54)$.

In the next set of experiments, we compare the basic stochastic version of our method, using estimators (25) — SNewton, the method with the variance reduction (Algorithm 4) — SVRNewton, and first-order algorithms (with constant step-size, tuned for each problem): SGD and SVRG [21]. We see (Figure 3) that using the variance reduction strategy significantly improve the convergence for both first-order and second-order stochastic optimization methods.

According to these graphs, our second-order algorithms can be more efficient when solving ill-conditioned problems, producing the better solution within a given computational time. See also Section E in the supplementary material for extra experiments.

## 8   Discussion

Let us discuss complexity estimates, which we established in our work. For the basic versions of our method we have the global convergence in the functional residual of the form

$$F(x_k) - F^* \quad \leq \quad \mathcal{O}\big(\tfrac{H_\nu D^{2+\nu}}{k^{1+\nu}}\big).$$

Note that the complexity parameter $H_\nu$ depends only on the variation of the Hessian (in arbitrary norm). It can be much smaller than the maximal eigenvalue of the Hessian, which typically appears in the rates of first-order methods. It is important that our algorithms are free from using the norms or any other particular parameters of the problem class.

At the same time, the arithmetic complexity of one step of our methods for simple sets can be estimated as the sum of the cost of computing the Hessian, and $\mathcal{O}(n^3)$ additional operations (to compute a suitable factorization of the matrix). For example, the cost of computing the gradient of Logistic Regression is $\mathcal{O}(Mn)$, and the Hessian is $\mathcal{O}(Mn^2)$, where $M$ is the dataset size. Hence, it is preferable to use our algorithms with exact steps in the situation when $M$ is much bigger than $n$.

## Broader Impact

This work does not present any foreseeable societal consequence.

## Acknowledgments and Disclosure of Funding

The research results of this paper were obtained in the framework of ERC Advanced Grant 788368.

## Footnotes

*Institute of Information and Communication Technologies, Electronics and Applied Mathematics (ICTEAM)

†Center for Operations Research and Econometrics (CORE)

[3]Indeed, it is enough to take $S_k(y) := \gamma_k \psi(x_k + \frac{1}{\gamma_k}(y - x_k))$.

[4] `https://www.csie.ntu.edu.tw/~cjlin/libsvmtools/datasets/`

[5] Clock time was evaluated using the machine with Intel Core i5 CPU, 1.6GHz; 8 GB RAM. All methods were implemented in C++. The source code can be found at `https://github.com/doikov/contracting-newton/`

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
