[Supplementary Material]

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

# Supplementary material

## A   Proof of Lemma 1

First, let us note that inequality (6) follows from the following simple observation, using Newton-Leibniz formula and Hölder continuity of the Hessian, for all $x, y \in \operatorname{dom} \psi$

$$\|\nabla f(y) - \nabla f(x) - \nabla^2 f(x)(y-x)\|_* = \|\int_0^1 (\nabla^2 f(x + \tau(y-x)) - \nabla^2 f(x))(y-x)d\tau\|_*$$

$$\overset{(5)}{\leq} \frac{H_\nu \|y-x\|^{1+\nu}}{1+\nu}.$$

We are ready to prove the lemma.

**Lemma 1**  *For all $x, y \in \operatorname{dom} \psi$ and $t \in [0,1]$, it holds*

$$f(y) \geq f(x) + \langle \nabla f(x), y-x \rangle + \tfrac{t}{2}\langle \nabla^2 f(x)(y-x), y-x \rangle - \frac{t^{1+\nu} H_\nu \|y-x\|^{2+\nu}}{(1+\nu)(2+\nu)}.$$

**Proof:**

Let us prove the following bound, for all $x, y \in \operatorname{dom} \psi$ and $t \in [0,1]$

$$\langle \nabla f(y) - \nabla f(x), y-x \rangle \geq t\langle \nabla^2 f(x)(y-x), y-x \rangle - \frac{t^{1+\nu} H_\nu \|y-x\|^{2+\nu}}{1+\nu}. \qquad (30)$$

For $t = 1$ it follows from (6). Therefore, we may assume that $t < 1$. Let us take $z_t := x + t(y-x)$. Then, by convexity of $f$, we have

$$\langle \nabla f(y), y-x \rangle = \tfrac{1}{1-t}\langle \nabla f(y), y-z_t \rangle$$

$$\geq \tfrac{1}{1-t}\langle \nabla f(z_t), y-z_t \rangle = \langle \nabla f(z_t), y-x \rangle.$$

Now, from Hölder continuity of the Hessian, we get

$$\langle \nabla f(z_t), y-x \rangle \overset{(6)}{\geq} \langle \nabla f(x), y-x \rangle + \langle \nabla^2 f(x)(z_t-x), y-x \rangle - \frac{H_\nu \|z_t-x\|^{1+\nu}\|y-x\|}{1+\nu}$$

$$= \langle \nabla f(x), y-x \rangle + t\langle \nabla^2 f(x)(y-x), y-x \rangle - \frac{t^{1+\nu} H_\nu \|y-x\|^{2+\nu}}{1+\nu}.$$

Thus we prove (30). Then, the claim of the lemma can be obtained by simple integration:

$$f(y) - f(x) - \langle \nabla f(x), y-x \rangle = \int_0^1 \langle \nabla f(z_\tau) - \nabla f(x), y-x \rangle d\tau$$

$$\overset{(30)}{\geq} \int_0^1 t\tau \langle \nabla^2 f(x)(y-x), y-x \rangle - \frac{(t\tau)^{1+\nu} H_\nu \|y-x\|^{2+\nu}}{1+\nu} d\tau$$

$$= \tfrac{t}{2}\langle \nabla^2 f(x)(y-x), y-x \rangle - \frac{t^{1+\nu} H_\nu \|y-x\|^{2+\nu}}{(1+\nu)(2+\nu)}.$$

$$\square$$

# B  Convergence of Contracting-Domain Newton Method

In this section, we prove the global convergence of Algorithms 1 and 2. We use the same notation as in the main part. There is a sequence of controlling coefficients $\{a_k\}_{k\geq 1}$ (see relations (12)), and a sequence of linear Estimating Functions $\{\phi_k(x)\}_{k\geq 0}$. We denote by $\mu \geq 0$ the constant of strong convexity of $\psi(\cdot)$. We allow $\mu = 0$ in the following auxiliary lemma, in order to cover both the general convex and the strongly convex cases.

**Lemma 2** *For the sequences $\{x_k\}_{k\geq 1}$ and $\{v_k\}_{k\geq 1}$, produced by Algorithm 1, we have*

$$A_k F(x_k) \quad \leq \quad \phi_k(x) + B_k(x), \qquad x \in \operatorname{dom}\psi, \tag{31}$$

*with*

$$B_k(x) \quad \equiv \quad \sum_{i=1}^{k} \left[ \frac{H_\nu a_i^{2+\nu} \|x - v_i\| \cdot \|x_{i-1} - v_i\|^{1+\nu}}{(1+\nu) A_i^{1+\nu}} - \frac{\mu a_i \|x - v_i\|^2}{2} - \frac{\mu a_i A_{i-1} \|x_{i-1} - v_i\|^2}{2 A_i} \right]. \tag{32}$$

**Proof:**

Let us prove (31) by induction. It obviously holds for $k = 0$, since $A_0 := 0$, $\phi_0(x) \equiv 0$, and $B_0(x) \equiv 0$ by definition. Assume that it holds for the current $k \geq 0$, and consider the next iterate. Stationary condition for the method step is

$$\langle \nabla f(x_k) + \nabla^2 f(x_k)(x_{k+1} - x_k), x - v_{k+1} \rangle + \psi(x) \quad \geq \quad \psi(v_{k+1}) + \tfrac{\mu}{2} \|x - v_{k+1}\|^2, \tag{33}$$

for all $x \in \operatorname{dom}\psi$. Then, we have

$$
\begin{aligned}
\phi_{k+1}(x) \quad &\equiv \quad a_{k+1}\big[ f(x_{k+1}) + \langle \nabla f(x_{k+1}), x - x_{k+1} \rangle + \psi(x) \big] + \phi_k(x) \\[4pt]
&\overset{(31)}{\geq} \quad a_{k+1}\big[ f(x_{k+1}) + \langle \nabla f(x_{k+1}), x - x_{k+1} \rangle + \psi(x) \big] + A_k F(x_k) - B_k(x) \\[4pt]
&\overset{(*)}{\geq} \quad A_{k+1}\big[ f(x_{k+1}) + \langle \nabla f(x_{k+1}), \tfrac{a_{k+1} x + A_k x_k}{A_{k+1}} - x_{k+1} \rangle \big] + a_{k+1}\psi(x) \\[4pt]
&\qquad + A_k \psi(x_k) - B_k(x) \\[4pt]
&= \quad A_{k+1} f(x_{k+1}) + a_{k+1}\langle \nabla f(x_{k+1}), x - v_{k+1} \rangle + a_{k+1}\psi(x) \\[4pt]
&\qquad + A_k \psi(x_k) - B_k(x) \\[4pt]
&= \quad A_{k+1} f(x_{k+1}) + a_{k+1}\big[ \langle \nabla f(x_k) + \nabla^2 f(x_k)(x_{k+1} - x_k), x - v_{k+1} \rangle + \psi(x) \big] \\[4pt]
&\qquad + a_{k+1}\langle \nabla f(x_{k+1}) - \nabla f(x_k) - \nabla^2 f(x_k)(x_{k+1} - x_k), x - v_{k+1} \rangle \\[4pt]
&\qquad + A_k \psi(x_k) - B_k(x) \\[4pt]
&\overset{(33),(6)}{\geq} \quad A_{k+1} f(x_{k+1}) + a_{k+1}\big[ \psi(v_{k+1}) + \tfrac{\mu}{2} \|x - v_{k+1}\|^2 \big] \\[4pt]
&\qquad - \frac{H_\nu a_{k+1}^{2+\nu} \|x - v_{k+1}\| \cdot \|v_{k+1} - x_k\|^{1+\nu}}{(1+\nu) A_{k+1}^{1+\nu}} + A_k \psi(x_k) - B_k(x) \\[4pt]
&\overset{(**)}{\geq} \quad A_{k+1} F(x_{k+1}) + \frac{\mu a_{k+1} \|x - v_{k+1}\|^2}{2} + \frac{\mu a_{k+1} A_k}{2 A_{k+1}} \|x_k - v_{k+1}\|^2 \\[4pt]
&\qquad - \frac{H_\nu a_{k+1}^{2+\nu} \|x - v_{k+1}\| \cdot \|v_{k+1} - x_k\|^{1+\nu}}{(1+\nu) A_{k+1}^{1+\nu}} + A_k \psi(x_k) - B_k(x) \\[4pt]
&\equiv \quad A_{k+1} F(x_{k+1}) - B_{k+1}(x),
\end{aligned}
$$

where $(*)$ and $(**)$ stand for convexity of $f$, and strong convexity of $\psi$, correspondingly. Thus we have (31) established for all $k \geq 0$. $\qquad\square$

## B.1 Proof of Theorem 1

**Theorem 1** *Let $A_k := k^3$, and consequently, $\gamma_k := 1 - \left(\frac{k}{k+1}\right)^3 = \mathcal{O}\left(\frac{1}{k}\right)$. Then for the sequence $\{x_k\}_{k\geq 1}$ generated by Algorithm 1, we have*

$$F(x_k) - F^* \leq \ell_k \stackrel{\text{def}}{=} F(x_k) - \frac{\phi_k^*}{A_k} \leq \mathcal{O}\left(\frac{H_\nu D^{2+\nu}}{k^{1+\nu}}\right).$$

**Proof:**

First, by convexity of $f$ we have, for all $x \in \operatorname{dom}\psi$

$$\phi_k(x) \leq A_k F(x).$$

Therefore, for the solution $x^*$ of our problem: $F^* = F(x^*)$, it holds

$$F(x_k) - F^* \leq F(x_k) - \frac{\phi_k(x^*)}{A_k} \leq \ell_k \stackrel{\text{def}}{=} F(x_k) - \frac{\phi_k^*}{A_k},$$

and this is the first part of (13).

At the same time, by Lemma 2, and using boundness of the domain, we have

$$\phi_k^* := \min_{x \in \operatorname{dom}\psi}\left\{\phi_k(x)\right\} \stackrel{(31)}{\geq} \min_{x \in \operatorname{dom}\psi}\left\{A_k F(x_k) - B_k(x)\right\}$$

$$\geq A_k F(x_k) - \frac{H_\nu D^{2+\nu}}{1+\nu}\sum_{i=1}^{k}\frac{a_i^{2+\nu}}{A_i^{1+\nu}}$$

Therefore, for the choice $A_k := k^3$, we finally obtain

$$\ell_k \leq \frac{H_\nu D^{2+\nu}}{(1+\nu)A_k}\sum_{i=1}^{k}\frac{a_i^{2+\nu}}{A_i^{1+\nu}} = \frac{H_\nu D^{2+\nu}}{(1+\nu)k^3}\sum_{i=1}^{k}\frac{(i^3-(i-1)^3)^{2+\nu}}{i^{3(1+\nu)}}$$

$$\leq \frac{H_\nu D^{2+\nu}}{(1+\nu)k^3}\sum_{i=1}^{k}\frac{3^{2+\nu}i^{2(2+\nu)}}{i^{3(1+\nu)}} = \frac{3^{2+\nu}H_\nu D^{2+\nu}}{(1+\nu)k^3}\sum_{i=1}^{k}i^{1-\nu}$$

$$= \mathcal{O}\left(\frac{H_\nu D^{2+\nu}}{k^{1+\nu}}\right).$$

$\square$

## B.2 Proof of Theorem 2

**Theorem 2** *Let $A_k := k^5$, and consequently, $\gamma_k := 1 - \left(\frac{k}{k+1}\right)^5 = \mathcal{O}\left(\frac{1}{k}\right)$. Then for the sequence $\{x_k\}_{k\geq 1}$ generated by Algorithm 1, we have*

$$F(x_k) - F^* \leq \ell_k \leq \mathcal{O}\left(\frac{H_\nu D^\nu}{\mu} \cdot \frac{H_\nu D^{2+\nu}}{k^{2+2\nu}}\right).$$

*Moreover, if the second-order condition number*

$$\omega_\nu \stackrel{\text{def}}{=} \left[\frac{H_\nu D^\nu}{(1+\nu)\mu}\right]^{\frac{1}{1+\nu}}$$

*is known, then, defining $A_k := (1 + \omega_\nu^{-1})^k$, $k \geq 1$, $A_0 := 0$, and $\gamma_k := \frac{1}{1+\omega_\nu}$, $k \geq 1$, $\gamma_0 := 1$, we obtain the global linear rate of convergence*

$$F(x_k) - F^* \leq \ell_k \leq \exp\left(-\frac{k-1}{1+\omega_\nu}\right) \cdot \frac{H_\nu D^{2+\nu}}{1+\nu}.$$

**Proof:**

Starting from the same reasoning, as in the proof of Theorem 1, we get

$$F(x_k) - F^* \leq \ell_k \stackrel{\text{def}}{=} F(x_k) - \frac{\phi_k^*}{A_k}.$$

Let us denote by $u_k$ the minimum of the Estimating Function $\phi_k$. Thus,

$$\ell_k \;=\; F(x_k) - \frac{\phi_k(u_k)}{A_k} \;\overset{(31)}{\leq}\; \frac{1}{A_k} B_k(u_k) \;\equiv\; \frac{1}{A_k} \sum_{i=1}^{k} B_k^{(i)},$$

with

$$
\begin{aligned}
B_k^{(i)} \;&\overset{\text{def}}{=}\; a_i\left[\frac{H_\nu a_i^{1+\nu}\|u_k-v_i\|\cdot\|x_{i-1}-v_i\|^{1+\nu}}{(1+\nu)A_i^{1+\nu}} - \frac{\mu\|u_k-v_i\|^2}{2}\right] - \frac{\mu a_i A_{i-1}\|x_{i-1}-v_i\|^2}{2A_i} \\[2mm]
&\leq\; a_i \max_{t\geq 0}\left\{\frac{H_\nu a_i^{1+\nu}\|x_{i-1}-v_i\|^{1+\nu}t}{(1+\nu)A_i^{1+\nu}} - \frac{\mu t^2}{2}\right\} - \frac{\mu a_i A_{i-1}\|x_{i-1}-v_i\|^2}{2A_i} \qquad (34)\\[2mm]
&=\; \frac{a_i}{2\mu}\left(\frac{H_\nu a_i^{1+\nu}\|x_{i-1}-v_i\|^{1+\nu}}{(1+\nu)A_i^{1+\nu}}\right)^2 - \frac{\mu a_i A_{i-1}\|x_{i-1}-v_i\|^2}{2A_i}.
\end{aligned}
$$

Therefore, for the choice $A_k := k^5$, we have

$$
\begin{aligned}
\ell_k \;&\leq\; \frac{1}{A_k}\sum_{i=1}^{k}\frac{a_i}{2\mu}\left(\frac{H_\nu a_i^{1+\nu}\|x_{i-1}-v_i\|^{1+\nu}}{(1+\nu)A_i^{1+\nu}}\right)^2 \;\leq\; \frac{H_\nu^2 D^{2(1+\nu)}}{2\mu(1+\nu)^2 A_k}\sum_{i=1}^{k}\frac{a_i^{2(1+\nu)+1}}{A_i^{2(1+\nu)}} \\[2mm]
&=\; \frac{H_\nu^2 D^{2(1+\nu)}}{2\mu(1+\nu)^2 k^5}\sum_{i=1}^{k}\frac{(i^5-(i-1)^5)^{2(1+\nu)+1}}{i^{10(1+\nu)}} \;\leq\; \frac{5^{2(1+\nu)+1}H_\nu^2 D^{2(1+\nu)}}{2\mu(1+\nu)^2 k^5}\sum_{i=1}^{k}i^{2-2\nu} \\[2mm]
&=\; \mathcal{O}\!\left(\frac{H_\nu D^\nu}{\mu}\cdot\frac{H_\nu D^{2+\nu}}{k^{2+2\nu}}\right).
\end{aligned}
$$

Thus we have justified (16). To obtain the linear rate (18), we set

$$A_k \;:=\; (1+\omega_\nu^{-1})^k, \qquad k\geq 1,$$

and $A_0 := 0$. So, $a_1 = A_1$ and

$$a_i \;=\; A_i - A_{i-1} \;=\; \omega_\nu^{-1}A_{i-1}, \qquad i\geq 2.$$

Therefore, for the values $\{B_k^{(i)}\}_{i=1}^{k}$, we have

$$B_k^{(1)} \;\leq\; a_1\frac{H_\nu D^{2+\nu}}{1+\nu} \;=\; A_1\frac{H_\nu D^{2+\nu}}{1+\nu},$$

and

$$
\begin{aligned}
B_k^{(i)} \;&\overset{(34)}{\leq}\; \frac{H_\nu^2 D^{2\nu}\|x_{i-1}-v_i\|^2 a_i^{3+2\nu}}{2\mu(1+\nu)^2 A_i^{2+2\nu}} - \frac{\mu a_i A_{i-1}\|x_{i-1}-v_i\|^2}{2A_i} \\[2mm]
&=\; \frac{\mu a_i A_{i-1}\|x_{i-1}-v_i\|^2}{2A_i}\left(\left[\frac{H_\nu D^\nu}{(1+\nu)\mu}\right]^2\frac{a_i^{2+2\nu}}{A_i^{1+2\nu}A_{i-1}} - 1\right) \\[2mm]
&\leq\; \frac{\mu a_i A_{i-1}\|x_{i-1}-v_i\|^2}{2A_i}\left(\left[\frac{H_\nu D^\nu}{(1+\nu)\mu}\right]^2\left[\frac{a_i}{A_{i-1}}\right]^{2(1+\nu)} - 1\right) \\[2mm]
&=\; 0, \qquad 2\leq i\leq k,
\end{aligned}
$$

since by our choice

$$\frac{a_i}{A_{i-1}} \;=\; \omega_\nu^{-1} \;\overset{(17)}{=}\; \left[\frac{(1+\nu)\mu}{H_\nu D^\nu}\right]^{\frac{1}{1+\nu}}.$$

Finally, we obtain

$$
\begin{aligned}
\ell_k \;&\leq\; \frac{1}{A_k}B_k^{(1)} \;\leq\; \frac{A_1}{A_k}\cdot\frac{H_\nu D^{2+\nu}}{1+\nu} \;=\; \frac{1}{(1+\omega_\nu^{-1})^{k-1}}\cdot\frac{H_\nu D^{2+\nu}}{1+\nu} \\[2mm]
&\leq\; \exp\!\left(-\frac{k-1}{1+\omega_\nu}\right)\cdot\frac{H_\nu D^{2+\nu}}{1+\nu}.
\end{aligned}
$$

$\square$

### B.3 Proof of Theorem 3

**Theorem 3** *Let $A_k := k^3$ and $\gamma_k := 1 - \left(\frac{k}{k+1}\right)^3 = \mathcal{O}\left(\frac{1}{k}\right)$. Then for the sequence $\{x_k\}_{k \geq 1}$ generated by Algorithm 2, we have*

$$F(x_k) - F^* \quad \leq \quad \ell_k \quad \leq \quad \mathcal{O}\left(\frac{H_\nu D^{2+\nu}}{k^{1+\nu}}\right).$$

**Proof:**

The proof is very similar to that one for Algorithm 1. First, stationary condition for one iteration of Algorithm 2 is

$$\langle \nabla f(x_k) + \nabla^2 f(x_k)(x_{k+1} - x_k), x - v_{k+1} \rangle + \frac{1}{\gamma_k} \psi\big(\gamma_k x + (1 - \gamma_k)x_k\big)$$

$$\geq \quad \frac{1}{\gamma_k} \psi(x_{k+1}), \tag{35}$$

for all $x \in \operatorname{dom} \psi$ and $k \geq 0$ (compare with (33)), where

$$v_{k+1} \quad := \quad x_k + \frac{1}{\gamma_k}(x_{k+1} - x_k) \quad \in \quad \operatorname{dom} \psi.$$

Now, let us prove by induction the following bound

$$\phi_k(x) \quad \geq \quad A_k F(x_k) - B_k, \qquad x \in \operatorname{dom} \psi, \tag{36}$$

with $B_k := \frac{H_\nu D^{2+\nu}}{1+\nu} \sum_{i=1}^{k} \frac{a_i^{2+\nu}}{A_i^{1+\nu}}$. It obviously holds for $k = 0$, since both sides are zero. Assume that it holds for the current $k \geq 0$. Then, we have for the next iterate

$$\phi_{k+1}(x) \quad \equiv \quad a_{k+1}\big[f(x_{k+1}) + \langle \nabla f(x_{k+1}), x - x_{k+1} \rangle + \psi(x)\big] + \phi_k(x)$$

$$\overset{(36)}{\geq} \quad a_{k+1}\big[f(x_{k+1}) + \langle \nabla f(x_{k+1}), x - x_{k+1} \rangle + \psi(x)\big] + A_k F(x_k) - B_k$$

$$\overset{(*)}{\geq} \quad A_{k+1}\big[f(x_{k+1}) + \langle \nabla f(x_{k+1}), \tfrac{a_{k+1}x + A_k x_k}{A_{k+1}} - x_{k+1} \rangle\big] + a_{k+1}\psi(x) + A_k\psi(x_k)$$

$$\quad - B_k$$

$$\overset{(**)}{\geq} \quad A_{k+1}\big[f(x_{k+1}) + \langle \nabla f(x_{k+1}), \tfrac{a_{k+1}x + A_k x_k}{A_{k+1}} - x_{k+1} \rangle + \psi\big(\tfrac{a_{k+1}x + A_k x_k}{A_{k+1}}\big)\big] - B_k,$$

where $(*)$ and $(**)$ stand for convexity of $f$ and $\psi$, correspondingly. Using both stationary condition and smoothness, we obtain, for all $x \in \operatorname{dom} \psi$

$$\langle \nabla f(x_{k+1}), \tfrac{a_{k+1}x + A_k x_k}{A_{k+1}} - x_{k+1} \rangle + \psi\big(\tfrac{a_{k+1}x + A_k x_k}{A_{k+1}}\big)$$

$$= \quad \gamma_k \langle \nabla f(x_{k+1}), x - v_{k+1} \rangle + \psi\big(\gamma_k x + (1 - \gamma_k)x_k\big)$$

$$= \quad \gamma_k \langle \nabla f(x_k) + \nabla^2 f(x_k)(x_{k+1} - x_k), x - v_{k+1} \rangle + \psi\big(\gamma_k x + (1 - \gamma_k)x_k\big)$$

$$\quad + \gamma_k \langle \nabla f(x_{k+1}) - \nabla f(x_k) - \nabla^2 f(x_k)(x_{k+1} - x_k), x - v_{k+1} \rangle$$

$$\overset{(35),(6)}{\geq} \quad \psi(x_{k+1}) - \tfrac{\gamma_k H_\nu \|x_{k+1} - x_k\|^{1+\nu}\|x - v_{k+1}\|}{1+\nu} \quad = \quad \psi(x_{k+1}) - \tfrac{\gamma_k^{2+\nu} H_\nu \|v_{k+1} - x_k\|^{1+\nu}\|x - v_{k+1}\|}{1+\nu}$$

$$\geq \quad \psi(x_{k+1}) - \tfrac{\gamma_k^{2+\nu} H_\nu D^{2+\nu}}{1+\nu}.$$

Therefore, we have

$$\phi_{k+1}(x) \quad \geq \quad A_{k+1}\big[f(x_{k+1}) + \psi(x_{k+1}) - \tfrac{\gamma_k^{2+\nu} H_\nu D^{2+\nu}}{1+\nu}\big] - B_k$$

$$= \quad A_{k+1}F(x_{k+1}) - B_{k+1},$$

and (36) is justified for all $k \geq 0$. Finally, by convexity of $f$, we get

$$
\begin{aligned}
F(x_k) - F^* \quad \leq \quad & \ell_k \;\overset{\text{def}}{=}\; F(x_k) - \tfrac{\phi_k^*}{A_k} \\[2mm]
&\overset{(36)}{\leq} \quad \tfrac{B_k}{A_k} \quad = \quad \tfrac{H_\nu D^{2+\nu}}{(1+\nu)A_k} \sum_{i=1}^{k} \tfrac{a_i^{2+\nu}}{A_i^{1+\nu}} \\[2mm]
&= \quad \mathcal{O}\big( \tfrac{H_\nu D^{2+\nu}}{k^{1+\nu}} \big),
\end{aligned}
$$

where the last equation holds from the choice $A_k := k^3$ (see the end of the proof of Theorem 1). $\square$

## C   Convergence of Aggregating Newton Method

In this section, we establish the convergence result for Algorithm 3.

### C.1   Proof of Theorem 4

**Theorem 4** *For the sequence $\{x_k\}_{k\geq 1}$ generated by Algorithm 3, relation (22) is satisfied.*

**Proof:**

Let us establish the relation (22) by induction. It obviously holds for $k = 0$. Assume that it is proven for the current iterate $k \geq 0$, and consider the next step:

$$
\begin{aligned}
&Q_{k+1}(v_{k+1}) \\[2mm]
&\equiv \quad a_{k+1}\big[ f(x_k) + \langle \nabla f(x_k), v_{k+1} - x_k \rangle + \tfrac{\gamma_k}{2}\langle \nabla^2 f(x_k)(v_{k+1} - x_k), v_{k+1} - x_k \rangle \\
&\qquad\qquad + \psi(v_{k+1}) \big] \;+\; Q_k(v_{k+1}) \\[2mm]
&\overset{(22)}{\geq} \quad a_{k+1}\big[ f(x_k) + \langle \nabla f(x_k), v_{k+1} - x_k \rangle + \tfrac{\gamma_k}{2}\langle \nabla^2 f(x_k)(v_{k+1} - x_k), v_{k+1} - x_k \rangle \\
&\qquad\qquad + \psi(v_{k+1}) \big] \;+\; A_k F(x_k) - \tfrac{C_k}{2} \\[2mm]
&= \quad A_{k+1}\big[ f(x_k) + \gamma_k\langle \nabla f(x_k), v_{k+1} - x_k \rangle + \tfrac{\gamma_k^2}{2}\langle \nabla^2 f(x_k)(v_{k+1} - x_k), v_{k+1} - x_k \rangle \big] \\
&\qquad\qquad + a_{k+1}\psi(v_{k+1}) + A_k \psi(x_k) - \tfrac{C_k}{2} \\[2mm]
&= \quad A_{k+1}\big[ f(x_k) + \langle \nabla f(x_k), x_{k+1} - x_k \rangle + \tfrac{1}{2}\langle \nabla^2 f(x_k)(x_{k+1} - x_k), x_{k+1} - x_k \rangle \big] \\
&\qquad\qquad + a_{k+1}\psi(v_{k+1}) + A_k \psi(x_k) - \tfrac{C_k}{2} \\[2mm]
&\overset{(7)}{\geq} \quad A_{k+1}\big[ f(x_{k+1}) - \tfrac{H_\nu \|x_{k+1} - x_k\|^{2+\nu}}{(1+\nu)(2+\nu)} \big] + a_{k+1}\psi(v_{k+1}) + A_k \psi(x_k) - \tfrac{C_k}{2} \\[2mm]
&= \quad A_{k+1}f(x_{k+1}) - \tfrac{A_{k+1}\gamma_k^{2+\nu} H_\nu \|v_{k+1} - x_k\|^{2+\nu}}{(1+\nu)(2+\nu)} + a_{k+1}\psi(v_{k+1}) + A_k \psi(x_k) - \tfrac{C_k}{2} \\[2mm]
&\geq \quad A_{k+1}f(x_{k+1}) - \tfrac{a_{k+1}\gamma_k^{1+\nu} \mathcal{H}_\nu D^{2+\nu}}{(1+\nu)(2+\nu)} + A_{k+1}\psi(x_{k+1}) - \tfrac{C_k}{2} \\[2mm]
&= \quad A_{k+1}F(x_{k+1}) - \tfrac{C_{k+1}}{2}.
\end{aligned}
$$

Thus, we have (22) justified for all $k \geq 0$. $\square$

## D  Convergence of stochastic methods

Let us consider the following general iterations, for solving optimization problem (1):

$$x_{k+1} \quad \in \quad \underset{y}{\mathrm{Argmin}} \Big\{ \langle g_k, y - x_k \rangle + \tfrac{1}{2} \langle H_k(y - x_k), y - x_k \rangle + S_k(y) \Big\}, \quad k \geq 0 \tag{37}$$

with $S_k(y) := \gamma_k \psi(x_k + \frac{1}{\gamma_k}(y - x_k))$. This is Algorithm 1 with substituted vector $g_k$ and matrix $H_k$ instead of the true gradient and the Hessian. First, we need to study the convergence of this process. For simplicity, let us study the case $\nu = 1$ only (convex functions with Lipschitz continuous Hessian, we denote the corresponding Lipschitz constant by $L_2$). Recall, that in this section we use the standard Euclidean norm for vectors and induced spectral norm for matrices.

As before, we use the sequence of positive numbers $\{a_k\}_{k \geq 1}$, and set

$$\gamma_k \quad := \quad \tfrac{a_{k+1}}{A_{k+1}}, \qquad A_k \quad \overset{\mathrm{def}}{=} \quad \sum_{i=1}^{k} a_i.$$

**Lemma 3** *For iterations* (37), *we have for all* $k \geq 1$

$$F(x_k) - F^* \quad \leq \quad \tfrac{B_k}{A_k}, \tag{38}$$

*with*

$$B_k \quad := \quad \tfrac{L_2 D^3}{2} \sum_{i=0}^{k-1} \tfrac{a_{i+1}^3}{A_{i+1}^2} + D \sum_{i=0}^{k-1} a_{i+1} \|\nabla f(x_i) - g_i\| + D^2 \sum_{i=0}^{k-1} \tfrac{a_{i+1}^2}{A_{i+1}} \|\nabla^2 f(x_i) - H_i\|.$$

**Proof:**

Let us prove by induction the following inequality

$$A_k F(x) \quad \geq \quad A_k F(x_k) - B_k, \qquad x \in \mathrm{dom}\, \psi. \tag{39}$$

It obviously holds for $k = 0$, and for $k \geq 1$ it is equivalent to (38).

Assume that (39) is satisfied for some $k \geq 0$, and consider the next step:

$$
\begin{aligned}
A_{k+1} F(x) \quad &= \quad a_{k+1} F(x) + A_k F(x) \\[2mm]
&\overset{(39)}{\geq} \quad a_{k+1} F(x) + A_k F(x_k) - B_k \\[2mm]
&\overset{(*)}{\geq} \quad A_{k+1} f\big(\tfrac{a_{k+1} x + A_k x_k}{A_{k+1}}\big) + a_{k+1} \psi(x) + A_k \psi(x_k) - B_k \\[2mm]
&\overset{(*)}{\geq} \quad A_{k+1} \big[ f(x_{k+1}) + \langle \nabla f(x_{k+1}), \tfrac{a_{k+1} x + A_k x_k}{A_{k+1}} - x_{k+1} \rangle \big] + a_{k+1} \psi(x) \\[2mm]
&\qquad + A_k \psi(x_k) - B_k,
\end{aligned}
\tag{40}
$$

where $(*)$ stands for convexity of $f$. Now, let us denote the point

$$v_{k+1} \quad := \quad x_k + \tfrac{1}{\gamma_k}(x_{k+1} - x_k) \quad \in \quad \mathrm{dom}\, \psi.$$

Then, stationary condition for the method step (37) can be written as

$$\langle g_k + H_k(x_{k+1} - x_k), x - v_{k+1} \rangle + \psi(x) \quad \geq \quad \psi(v_{k+1}), \tag{41}$$

for all $x \in \operatorname{dom} \psi$. Therefore,

$$A_{k+1}\langle \nabla f(x_{k+1}), \tfrac{a_{k+1}x + A_k x_k}{A_{k+1}} - x_{k+1} \rangle + a_{k+1}\psi(x)$$

$$= \quad a_{k+1}\big[ \langle \nabla f(x_{k+1}), x - v_{k+1} \rangle + \psi(x) \big]$$

$$= \quad a_{k+1}\big[ \langle g_k + H_k(x_{k+1} - x_k), x - v_{k+1} \rangle + \psi(x)$$

$$+ \langle \nabla f(x_k) - g_k, x - v_{k+1} \rangle$$

$$+ \langle (\nabla^2 f(x_k) - H_k)(x_{k+1} - x_k), x - v_{k+1} \rangle$$

$$+ \langle \nabla f(x_{k+1}) - \nabla f(x_k) - \nabla^2 f(x_k)(x_{k+1} - x_k), x - v_{k+1} \rangle \big] \qquad (42)$$

$$\overset{(41),(6)}{\geq} \quad a_{k+1}\big[ \psi(v_{k+1}) - \|\nabla f(x_k) - g_k\| \cdot \|x - v_{k+1}\|$$

$$- \gamma_k \|\nabla^2 f(x_k) - H_k\| \cdot \|v_{k+1} - x_k\| \cdot \|x - v_{k+1}\|$$

$$- \tfrac{L_2 \gamma_k^2 \|v_{k+1} - x_k\|^2 \cdot \|x - v_{k+1}\|}{2} \big]$$

$$\geq \quad a_{k+1}\psi(v_{k+1}) - a_{k+1} D \|\nabla f(x_k) - g_k\|_* - \tfrac{a_{k+1}^2 D^2 \|\nabla^2 f(x_k) - H_k\|}{A_{k+1}} - \tfrac{a_{k+1}^3 L_2 D^3}{A_{k+1}^2}.$$

Thus, combining all together, and using convexity of $\psi$, we obtain

$$A_{k+1} F(x) \overset{(40),(42)}{\geq} \quad A_{k+1} f(x_{k+1}) + a_{k+1}\psi(v_{k+1}) + A_k \psi(x_k) - B_k$$

$$- a_{k+1} D \|\nabla f(x_k) - g_k\| - \tfrac{a_{k+1}^2 D^2 \|\nabla^2 f(x_k) - H_k\|}{A_{k+1}} - \tfrac{a_{k+1}^3 L_2 D^3}{A_{k+1}^2}$$

$$\geq \quad A_{k+1} F(x_{k+1}) - B_{k+1}.$$

So, we have (39) justified for all $k \geq 0$. $\qquad\qquad\square$

Now, we are ready to prove convergence results for the process (37) with the basic variant of stochastic estimators (25), and with the variance reduction strategy for the gradients, incorporated into Algorithm 4.

## D.1 Proof of Theorem 5

**Theorem 5** *Let each component $f_i(\cdot)$ be Lipschitz continuous on $\operatorname{dom} \psi$ with constant $L_0$, and have Lipschitz continuous gradients and Hessians on $\operatorname{dom} \psi$ with constants $L_1$ and $L_2$, respectively. Let $\gamma_k := 1 - \left(\tfrac{k}{k+1}\right)^3 = \mathcal{O}\left(\tfrac{1}{k}\right)$. Set*

$$m_k^g \quad := \quad 1/\gamma_k^4, \qquad m_k^H \quad := \quad 1/\gamma_k^2.$$

*Then, for the iterations $\{x_k\}_{k \geq 1}$ of Algorithm (1), based on estimators (25), it holds*

$$\mathbb{E}[F(x_k) - F^*] \quad \leq \quad \mathcal{O}\left( \tfrac{L_2 D^3 + L_1 D^2 (1 + \log(n)) + L_0 D}{k^2} \right).$$

**Proof:**

Let us fix iteration $k \geq 0$. For one uniform random sample $i \in \{1, \ldots, M\}$, we have

$$\mathbb{E}\|\nabla f(x_k) - \nabla f_i(x_k)\|^2 \quad = \quad \mathbb{E}\|\nabla f_i(x_k)\|^2 - \|\nabla f(x_k)\|^2 \quad \leq \quad L_0^2. \qquad (43)$$

Therefore, for the random batch of size $m_k^g$, we obtain

$$
\begin{aligned}
\mathbb{E}\|\nabla f(x_k) - g_k\| &\leq \sqrt{\mathbb{E}\|\nabla f(x_k) - g_k\|^2} \\
&= \sqrt{\frac{1}{(m_k^g)^2}\mathbb{E}\|\sum_{i \in S_k^g}(\nabla f(x_k) - \nabla f_i(x_k))\|^2} \\
&= \sqrt{\frac{1}{(m_k^g)^2}\sum_{i \in S_k^g}\mathbb{E}\|\nabla f(x_k) - \nabla f_i(x_k)\|^2} \\
&\overset{(43)}{\leq} \frac{L_0}{\sqrt{m_k^g}}.
\end{aligned}
\tag{44}
$$

More advanced reasoning for matrices (Matrix Bernstein Inequality; see Chapter 6 in [40]) gives

$$
\begin{aligned}
\mathbb{E}\|\nabla^2 f(x_k) - H_k\| &\leq L_1\left(\sqrt{\frac{2\log(2n)}{m_k^H}} + \frac{2\log(2n)}{3m_k^H}\right) \\
&\leq \frac{L_1(3\sqrt{2\log(2n)} + 2\log(2n))}{3\sqrt{m_k^H}} \leq \frac{L_1(6 + 7\log(2n))}{6\sqrt{m_k^H}}.
\end{aligned}
\tag{45}
$$

So, using these estimates together, we have, for every $k \geq 1$

$$
\begin{aligned}
\mathbb{E}[F(x_k) - F^*] &\overset{(38)}{\leq} \frac{1}{A_k}\left(\frac{L_2 D^3}{2}\sum_{i=0}^{k-1}\frac{a_{i+1}^3}{A_{i+1}^2} + D\sum_{i=0}^{k-1}a_{i+1}\mathbb{E}\|\nabla f(x_i) - g_i\| \right. \\
&\qquad\qquad \left. + D^2\sum_{i=0}^{k-1}\frac{a_{i+1}^2}{A_{i+1}}\mathbb{E}\|\nabla^2 f(x_i) - H_i\|\right) \\
&\overset{(44),(45)}{\leq} \frac{1}{A_k}\left(\frac{L_2 D^3}{2}\sum_{i=0}^{k-1}\frac{a_{i+1}^3}{A_{i+1}^2} + L_0 D\sum_{i=0}^{k-1}\frac{a_{i+1}}{\sqrt{m_i^g}}\right. \\
&\qquad\qquad \left. + \frac{L_1 D^2(6 + 7\log(2n))}{6}\sum_{i=0}^{k-1}\frac{a_{i+1}^2}{A_{i+1}\sqrt{m_i^H}}\right) \\
&\overset{(26)}{=} \frac{1}{A_k}\left(\frac{L_2 D^3}{2} + L_0 D + \frac{L_1 D^2(6 + 7\log(2n))}{6}\right)\sum_{i=0}^{k-1}\frac{a_{i+1}^3}{A_{i+1}^2}.
\end{aligned}
$$

Thus, for the choice $A_k := k^3$, we get

$$
\mathbb{E}[F(x_k) - F^*] \leq \mathcal{O}\left(\frac{L_2 D^3 + L_1 D^2(1 + \log(n)) + L_0 D}{k^2}\right).
$$

$\square$

## D.2 Proof of Theorem 6

**Theorem 6** *Let each component $f_i(\cdot)$ have Lipschitz continuous gradients and Hessians on $\operatorname{dom}\psi$ with constants $L_1$ and $L_2$, respectively. Let $\gamma_k := 1 - \left(\frac{k}{k+1}\right)^3 = \mathcal{O}(\frac{1}{k})$. Set batch size*

$$
m_k := 1/\gamma_k^2.
$$

*Then, for all iterations $\{x_k\}_{k \geq 1}$ of Algorithm 4, we have*

$$
\mathbb{E}[F(x_k) - F^*] \leq \mathcal{O}\left(\frac{L_2 D^3 + L_1 D^2(1 + \log(n)) + L_1^{1/2}D(F(x_0) - F^*)}{k^2}\right).
$$

**Proof:**

Let us consider the following stochastic estimate

$$
g_k^i := \nabla f_i(x_k) - \nabla f_i(z_k) + \nabla f(z_k),
$$

for a uniform random sample $i \in \{1, \ldots, M\}$, and a current iterate $k \geq 0$. We denote by $x^*$ the solution of our problem: $F^* = F(x^*)$, stationary condition for which is

$$\langle \nabla f(x^*), x - x^* \rangle + \psi(x) \geq \psi(x^*), \qquad x \in \operatorname{dom} \psi. \tag{46}$$

Then, it holds

$$
\begin{aligned}
\mathbb{E} \|\nabla f(x_k) - g_k^i\|^2 &= \mathbb{E} \| (\nabla f(x_k) - \nabla f(x^*)) \\
&\quad + (\nabla f_i(z_k) - \nabla f_i(x^*) - \nabla f(z_k) + \nabla f(x^*)) \\
&\quad + (\nabla f_i(x^*) - \nabla f_i(x_k)) \|^2 \\
&\leq 3\mathbb{E} \|\nabla f(x_k) - \nabla f(x^*)\|^2 \\
&\quad + 3\mathbb{E} \| (\nabla f_i(z_k) - \nabla f_i(x^*)) - (\nabla f(z_k) - \nabla f(x^*)) \|^2 \\
&\quad + 3\mathbb{E} \|\nabla f_i(x_k) - \nabla f_i(x^*)\|^2 \\
&\leq 3 \Big( \mathbb{E}\|\nabla f(x_k) - \nabla f(x^*)\|^2 + \mathbb{E}\|\nabla f_i(z_k) - \nabla f_i(x^*)\|^2 \\
&\quad + \mathbb{E}\|\nabla f_i(x_k) - \nabla f_i(x^*)\|^2 \Big),
\end{aligned}
$$

where we used the following simple bounds:

$$
\begin{aligned}
\|a + b + c\|^2 &\leq 3\|a\|^2 + 3\|b\|^2 + 3\|c\|^2, \\
\mathbb{E}\|\xi - \mathbb{E}\xi\|^2 &\leq \mathbb{E}\|\xi\|^2,
\end{aligned}
$$

which are valid for any $a, b, c \in \mathbb{R}^n$ and arbitrary random vector $\xi \in \mathbb{R}^n$.

Now, by Lipschitz continuity of the gradients, we have (see Theorem 2.1.5 in [31])

$$
\begin{aligned}
\|\nabla f(x_k) - \nabla f(x^*)\|^2 &\leq 2L_1 \big( f(x_k) - f(x^*) - \langle \nabla f(x^*), x_k - x^* \rangle \big) \\
&\overset{(46)}{\leq} 2L_1 \big( F(x_k) - F^* \big).
\end{aligned}
$$

The same holds for the random sample $i$, for arbitrary fixed $x \in \operatorname{dom} \psi$

$$
\begin{aligned}
\mathbb{E}_i \|\nabla f_i(x) - \nabla f_i(x^*)\|^2 &\leq 2L_1 \mathbb{E}_i \big[ f_i(x) - f_i(x^*) - \langle \nabla f_i(x^*), x - x^* \rangle \big] \\
&= 2L_1 \big( f(x) - f(x^*) - \langle \nabla f(x^*), x - x^* \rangle \big) \\
&\overset{(46)}{\leq} 2L_1 \big( F(x) - F^* \big).
\end{aligned}
$$

Thus, we obtain

$$\mathbb{E}\|\nabla f(x_k) - g_k^i\|^2 \leq 12 L_1 \mathbb{E}[F(x_k) - F^*] + 6 L_1 \mathbb{E}[F(z_k) - F^*]. \tag{47}$$

Consequently, for the random batch

$$g_k := \frac{1}{m_k} \sum_{i \in S_k} g_k^i,$$

we have (compare with (44))

$$
\begin{aligned}
\mathbb{E}\|\nabla f(x_k) - g_k\| &\leq \sqrt{\frac{1}{(m_k)^2} \sum_{i \in S_k} \mathbb{E}\|\nabla f(x_k) - g_k^i\|^2} \\
&\overset{(47)}{\leq} \sqrt{\frac{6L_1}{m_k} \big( 2\mathbb{E}[F(x_k) - F^*] + \mathbb{E}[F(z_k) - F^*] \big)} \\
&\leq \sqrt{\frac{12L_1}{m_k} \mathbb{E}[F(x_k) - F^*]} + \sqrt{\frac{6L_1}{m_k} \mathbb{E}[F(z_k) - F^*]}.
\end{aligned}
\tag{48}
$$

So, using the variance reduction for the gradients, and the basic estimate for the Hessians, we have, for every $k \geq 1$

$$\mathbb{E}[F(x_k) - F^*] \overset{(38),(48),(45)}{\leq} \frac{1}{A_k}\left(\frac{L_2 D^3}{2}\sum_{i=0}^{k-1}\frac{a_{i+1}^3}{A_{i+1}^2}\right.$$

$$+ D\sqrt{6L_1}\sum_{i=0}^{k-1}\frac{a_{i+1}}{\sqrt{m_i}}\left(\sqrt{2\mathbb{E}[F(x_i)-F^*]}+\sqrt{\mathbb{E}[F(z_i)-F^*]}\right)$$

$$\left.+ \frac{L_1 D^2(6+7\log(2n))}{6}\sum_{i=0}^{k-1}\frac{a_{i+1}^2}{A_{i+1}\sqrt{m_i}}\right)$$

$$\overset{(28)}{=} \frac{1}{A_k}\left(\left[\frac{3L_2 D^3 + L_1 D^2(6+7\log(2n))}{6}\right]\sum_{i=0}^{k-1}\frac{a_{i+1}^3}{A_{i+1}^2}\right.$$

$$\left.+ D\sqrt{6L_1}\sum_{i=0}^{k-1}\frac{a_{i+1}^2}{A_{i+1}}\left(\sqrt{2\mathbb{E}[F(x_i)-F^*]}+\sqrt{\mathbb{E}[F(z_i)-F^*]}\right)\right).$$

Now, let us set $A_{i+1} := (i+1)^3$, and thus $a_{i+1} := (i+1)^3 - i^3 \leq 3(i+1)^2$, so we have

$$\mathbb{E}[F(x_k) - F^*] \leq \frac{\alpha + \beta(\sqrt{2}+1)(F(x_0)-F^*)}{k^2}$$

$$+ \frac{\beta}{k^3}\sum_{i=1}^{k-1}\left((i+1)\left(\sqrt{2\mathbb{E}[F(x_i)-F^*]}+\sqrt{\mathbb{E}[F(z_i)-F^*]}\right)\right), \qquad (49)$$

where

$$\alpha := 27 \cdot \left[\frac{3L_2 D^3 + L_1 D^2(6+7\log(2n))}{6}\right], \qquad \beta := 9 \cdot D\sqrt{6L_1}.$$

We are going to prove by induction, for every $k \geq 1$

$$\mathbb{E}[F(x_k) - F^*] \leq \frac{c}{k^2}, \qquad (50)$$

with

$$c := \left(4\beta + \sqrt{\alpha + 3\beta(F(x_0)-F^*) + 16\beta^2}\right)^2 \leq 74\beta^2 + 2\alpha + 6\beta(F(x_0)-F^*) \qquad (51)$$

$$= \mathcal{O}\left(L_2 D^3 + L_1 D^2(1+\log(n)) + L_1^{1/2}D(F(x_0)-F^*)\right).$$

Hence, if (50) is true, then we essentially obtain the claim of the theorem. For $k=1$, (50) follows directly from (49). Assume that (50) holds for all $1 \leq i \leq k$, and consider iteration $k+1$:

$$\mathbb{E}[F(x_{k+1})-F^*] \overset{(49),(50)}{\leq} \frac{\alpha+\beta(\sqrt{2}+1)(F(x_0)-F^*)}{k^2} + \frac{\beta}{k^3}\sum_{i=1}^{k}\left((i+1)\left(\frac{\sqrt{2c}}{i}+\frac{\sqrt{c}}{\pi(i)}\right)\right)$$

$$\overset{(*)}{\leq} \frac{\alpha+\beta(\sqrt{2}+1)(F(x_0)-F^*)}{k^2} + \frac{\beta\sqrt{c}}{k^3}\sum_{i=1}^{k}\left((i+1)\left(\frac{2\sqrt{2}+4}{i+1}\right)\right)$$

$$= \frac{\alpha+(\sqrt{2}+1)\beta(F(x_0)-F^*)+(2\sqrt{2}+4)\beta\sqrt{c}}{k^2}$$

$$\leq \frac{\alpha+3\beta(F(x_0)-F^*)+8\beta\sqrt{c}}{k^2} \overset{(51)}{=} \frac{c}{k^2},$$

where in $(*)$ we have used two simple bounds: $i \leq 2\pi(i)$, and $i+1 \leq 2i$, valid for all $i \geq 1$. $\qquad \square$

# E  Extra experiments

In this section, we provide additional experimental results for the problem of training Logistic Regression model, regularized by $\ell_2$-ball constraints: Figure 4 for the exact methods, and Figure 6 for the stochastic algorithms.

**Figure 4:** Training logistic regression, datasets: *a9a* ($M = 32561, n = 123$), *connect-4* ($M = 67557, n = 126$), *mnist* ($M = 60000, n = 780$).

We see, that the second-order schemes usually outperforms first-order methods, in terms of the number of iterations, and the number of epochs. Despite the fact, that the Newton step is more expensive, in many situations we see superiority of the second-order schemes in terms of the total computational time as well.

Comparing Contracting-Domain Newton Method (Algorithm 1), and Aggregating Newton Method (Algorithm 3), we conclude that both of the algorithms show reasonably good performance in practice. The latter one works a bit slower. However, the aggregation of the Hessians helps to improve numerical stability. On Figure 5, we demonstrate influence of the parameter of inner accuracy (EPS), which we use in our subsolver, on the convergence of the algorithms. We see much more robust behaviour for Aggregating Newton Method, while the first algorithm can potentially stop, or even start to diverge, if the parameter is chosen in a wrong way.

To compute one step of our second-order methods for this task, we need to solve subproblem (20) for $p = 2$. This is minimization of quadratic function over the standard Euclidean ball. First, we compute *tridiagonal* decomposition of the Hessian (it requires $\mathcal{O}(n^3)$ arithmetical operations). Then, we solve the dual to our subproblem (which is maximization of one-dimensional concave function) by classical Newton iterations (the cost of each iteration is $\mathcal{O}(n)$). For more details, see Chapter 7 in [9].

**Figure 5:** Influence of the parameter of inner accuracy.

**Figure 6:** Stochastic methods for training logistic regression, datasets: *mnist* ($M = 60000, n = 780$), *YearPredictionMSD* ($M = 463715, n = 90$), *HIGGS2m* ($M = 2 \cdot 10^6, n = 28$).