[Reviews · NeurIPS 2020]

Review 1

Summary and Contributions: This paper presents new second-order algorithms for the prototypical composite convex optimization problem. First, the paper introduces a new global second-order lower approximation. Based on this lower approximation model, the paper introduces a new second-order optimization algorithm. The proposed method successively minimizes the lower approximation model of the smooth term augmented by the nonsmooth term and constructs the solution to the original problem as a convex combination of the solutions to these subproblems. In this regard, it can be seen as a second-order modification of the Frank-Wolfe method, which considers a second-order lower model instead of the first-order. The analysis establishes O(1/k^{1+v}) convergence rate, where v in [0,1] is the Holder continuity degree of the Hessian. When the composite component is strongly convex, this rate becomes O(1/k^{2+2v}). The proposed method is universal, in the sense that it adapts to the unknown Holder continuity degree and constant. Then, à la dual averaging gradient methods, the paper presents another second-order optimization algorithm, based on an aggregated model which “accumulates second-order information into quadratic Estimation Functions.” The main advantage of this second approach is that the new information enters into the model with increasing weights. The paper extends these methods to the stochastic optimization setting. While the direct extension has O(k^4) sample complexity with O(1/k^2) convergence rate, the sample complexity is reduced to O(k^2) using the variance reduction techniques. The paper also provides some interpretations of the proposed methods (Frank-Wolfe, trust-region, dual averaging, etc.) and demonstrates computational results for the logistic regression problem with the l2-ball constraint. -- update after author response: I have read all the reviews and the authors' feedback carefully and took it into account in my final evaluation.

Strengths: The novelty of the proposed method and the quality of the presentation are the major two strengths of the paper. In detail, the proposed method is universal, so it does not require the knowledge of the Holder continuity degree and constant of the Hessian but gets the optimal rates. It has direct extensions for strongly-convex and stochastic problems. The proposed method is also affine invariant, so it can be practically appealing when dealing with ill-conditioned problems.

Weaknesses: The paper lacks a detailed discussion on the potential applications and the significance of the proposed approach. I do not see a clear answer to the questions “for which problems the proposed approach has practical benefits?” or “what is the potential impact of the proposed approach?” The paper considers the logistic regression with l2-ball constrained in the numerical experiments section. The proposed approach is more efficient than the baseline considered in these experiments for reaching high-accuracy in terms of functional residual. Its impact is less clear for low-accuracy approximations, which is practically more interesting for the considered logistic regression problem.

Correctness: The claims and the method are correct. I have read the main text carefully and did not encounter any problem. I did not proof-read the supplementary material but I skimmed through it. The claimed results are reasonable. The code is also provided for the reproducibility of the empirical results, but I did not check the code.

Clarity: The paper is very well written. The presentation is very scholarly, there are no over-claims. The title descriptive, the abstract is concise but complete. From the optimization perspective, the motivation is clear (maybe it is less so from the practical perspective, see my comments in the Weaknesses section). The mathematical writing is also high quality, notation is simple and consistent throughout the text. It is easy to follow the ideas and the proofs due to this high quality of presentation. I enjoyed reading the paper.

Relation to Prior Work: The paper presents a good overview of the related work in the very beginning in the introduction (paragraphs 2-3). Starting from paragraph 4, the paper describes the contributions and the difference of the proposed approach from the state of the art. Some technical connections/similarities with the existing work are discussed when appropriate.

Reproducibility: Yes

Additional Feedback: - Can you provide some details on the subsolver that you used in the experiments? (maybe in the supplements) - Linguistic@line177: “less or equal than k” -> less than or equal to k - Can you explain how did you initialize algorithms in numerical experiments? In particular, for many of the problems FW seems to finish at a worse estimation than the initial point after a few thousands of iterations (Figure 4, 5). Why does this happen? Would FW with line-search be a better competitor?


Review 2

Summary and Contributions: This paper reexamines the convergence rate of second-order generalization of the conditional gradient methods for composite convex optimizations. It improves the previously known convergence rate for this type of algorithms. It also proposes a novel version of these algorithms (referred to as Aggregating Newton Method) as well as a stochastic version of these algorithms for finite sum minimization. It conducts various numerical experiments using the Logistic Regression model. =========Update========================================= My concerns have been addressed in the rebuttal. I kept my score as it is.

Strengths: It provides a comprehensive review of previous literature on the topic. The convergence analysis is an improvement to previously established bounds for these types of algorithms. It also proposes a more advanced version of the second-order conditional gradient method. The mathematical development for the establishment of the convergence analysis is thorough. It also studies the stochastic version of these algorithms for finite sum minimization and can inspire future research in the field. The experiment section has been executed very nicely and provides enough practical insights.

Weaknesses: Based on their experiments, it seems that the previously proposed Contracting Trust-Region (Ref [29] or Algorithm 2 of this paper ) works better than their novel proposal of Aggregating Newton Method.

Correctness: The theoretical and empiracl results seem to be correct and consistent with previous research in the field.

Clarity: Yes.

Relation to Prior Work: It has an impressive review of previous research in the field.

Reproducibility: Yes

Additional Feedback: It is hard to track when and when the composite term is free of the strong convexity assumption. Since ref [29] is the closest work to this paper, it could be beneficial to explain the contributions of [29] more clear. Line 171: It is interesting, that > redundant comma


Review 3

Summary and Contributions: This paper introduces a new second-order algorithm to solve convex composite optimization problems. Assuming that the Hessian is \nu-Holder continuous, their algorithm achieves a convergence rate O(1/k^{1+\nu}). Importantly, it is universal (independent of the knowledge of \nu and other function parameters) and it has a computable accuracy certificate. In contrast to other existing second-order methods which are based on cubic regularization and which have similar complexity, this new algorithm has the affine-invariance property. The authors provide several variants of their method. When the composite component is \mu-strongly convex, there is a universal variant of the algorithm with convergence rate O(1/k^{2+2\nu}) and a non-universal variant with linear convergence rate. The authors provide a second variant, which already exists in the literature, but they improve the analysis of the convergence rate (going from O(1/k) to O(1/k^2) for Lipschitz continuous Hessians). Finally, the authors provide an improvement which aggregates past information and achieves a rate O(1/k^{2+\nu}), at the expense of the accuracy certificate (which needs to be estimated). They also consider empirical risk minimization, and provide a variance reduction based algorithm with convergence rate O(1/k^2) provided that the batch sizes are O(k^2) for the gradients and Hessians.

Strengths: The theoretical results of the paper are strong and interesting. The main contribution is to provide an affine-invariant second-order method for solving this class of composite optimization problems, as well as to develop new analysis techniques to improve the convergence rates of an existing algorithm.

Weaknesses: The stochastic variants of the method for empirical risk minimization could be discussed in more details. The first variant is based on Algorithm 1 whereas the second variant (Algorithm 4) is based on Algorithm 2. In the deterministic case, Algorithms 1 and 2 are compared based on the improved convergence rate of Algorithm 1 when the composite component is strongly convex. A comparison of these two methods would be interesting in the stochastic case. The advantage of the proposed method over simple non-affine invariant methods (e.g., first order methods for composite optimization) is not clear. The authors should discuss the overall complexity, since the proposed method is more expensive per iteration.

Correctness: The theoretical claims are correct. The empirical methodology is fair. It would be great to provide more details regarding the numerical simulations, especially for the evaluation of the stochastic algorithms (e.g., average performance and standard deviations). Further, some additional discussion regarding the empirical performance in light of the theoretical convergence rates would be useful. For instance, on Figure 2, according to Eq. (13) and (23), I would expect a priori Aggr. Newton to be faster than Contr. Newton (in terms of iteration count). Similarly, on Figure 3, according to Eq. (27) and (29), I would expect a priori SNewton and SVRNewton to behave quite similarly. I am also wondering about the average empirical performance of these two methods (Figure 3 seems to display a single run of each method).

Clarity: The paper is well-written. It is pleasant to the reader to follow the theoretical and algorithmic developments.

Relation to Prior Work: Yes. The related literature is well described, and the paper’s novelties are easily identifiable to the reader.

Reproducibility: Yes

Additional Feedback: I have read the rebuttal, which addressed my main concerns. In particular, the authors clarified that the parameter H_ν in the convergence rates has a better dependence to Hessian ill-conditioning compared to first order methods. The authors could discuss this point and computational tradeoffs with respect to iteration/total complexity in the main paper. I will keep my original score (7).


Review 4

Summary and Contributions: New second order algorithms, named as contracting domain Newton methods, for composite convex optimization are developed based on global second order lower approximation for the smooth component of the objective.

Strengths: The strengths of the paper are as follows. 1. New algorithms are developed with rigorous proofs. 2. The convergence of the algorithms is O(1/k^2) for convex functions with Lipschitz continuous Hessian. 3. The new method is affine invariant and universal, i.e., it does not depend on the norms and parameters of the problem class. This is a very nice point. 4. The authors present an optimization process called Aggregating Newton Method with the convergence rate of O(1/k^{1+v}) for general convex case. 4. Experiments have been done to support the theory.

Weaknesses: The weaknesses of the paper are as follows. 1. It seems that the main assumption -- bounded D -- is very strong. If it is like that, then the received results in the paper might be weak compared to the previous ones (Cubic Newton). 2. It seems that the new algorithms do not yield a better convergence rate compared to the previous results as mentioned in the paper.

Correctness: It seems all the claims are correct.

Clarity: Yes, it is. The paper is very well written.

Relation to Prior Work: Yes, it is.

Reproducibility: Yes

Additional Feedback:


Review 5

Summary and Contributions: The authors try to present a new variant of second-order algorithm for convex optimization, named Contracting-domain Newton methods. With Lipschitz continuous Hessian, the authors try to prove a 1/k^2 global convergence as theoretical foundation for correctness and contributions. The author also tries to include stochastic variants of the algorithm with proven theoretical results.

Strengths: If the statement is correct with any \nu \in [0,1], the contributions to the optimization community will be novel as it provides a new routine for developing second-order optimization methods. However, the author is claim for some \nu in [0,1] which largely weakens the contributions. The paper is highly related to topics in NeurIPS community as an important topic in optimization.

Weaknesses: However, the author only asks the Holder condition to be held for some \nu. This is of course true under the case of Nesterov's cubic regularization. For \nu=0, it actually does not even hold for some common nonlinear functions. The only obvious contributions should be the case when \nu \in (0,1).

Correctness: The proof looks correct. However, I have one questions about the paper. The authors state that based on Taylor's approximation and Holder's condition, (6) holds true. However, the general Taylor's expansion is based on an integer exponent n, and it is quite important to prove it for general \nu\in(0,1) for (6). I think it would be more clear if the authors can also provide details to prove (6). Otherwise, it is not obvious. Update: the authors addressed this issue in the response. it's better to include the steps with Newton-Leibniz formula as one sentence in the main context or in the supplementary materials.

Clarity: The paper is somewhat well written but it misses a good conclusion with discussions. Update: still missing a good conclusion?

Relation to Prior Work: Except for some second-order algorithms, the authors also mentions some related first-order algorithms in convex optimization, e.g. SAG[36], SAGA[9], SVRG[15, 19, 20]; however, several important works are still missing, e.g. Katyusha: the first direct acceleration of stochastic gradient methods, STOC 2017. SARAH: A Novel Method for Machine Learning Problems Using Stochastic Recursive Gradient, ICML 2017. Update: the authors addressed this issue.

Reproducibility: Yes

Additional Feedback:


Review 6

Summary and Contributions: This paper studies the global convergence theory of second-order algorithms for composite convex optimization. The authors propose a novel type of algorithms, called contracting-domain Newton methods, which are based on a new global second-order approximation. The empirical studies show the proposed algorithms is better than baseline algorithms.

Strengths: I believe this is a high quality paper, which provides several interesting observation for second order optimization. The proposed contracting-domain Newton methods enjoy the following nice properties: 1. The algorithm is based on a tighter second-order lower bound approximation, which leads to these methods has global convergence guarantee. The algorithm can be interpreted as second-order generalization of conditional gradient methods or as a variant of trust-region method. 2. The convergence rate of proposed aggregated Newton method is no worse than cubic Newton, but each iteration do not require to solve a more complex cubic regularization problem. 3. For stochastic finite-sum minimization, contracting-domain Newton can be easily extended to (variance-reduced) stochastic algorithms, which is more efficient for large scale optimization problem in machine learning. 4. All experimental results validates the effectiveness of the proposed algorithms.

Weaknesses: 1. It is prefer to empirically compare proposed algorithms with existing second-order methods with line-search. 2. Line 51 states aggregating Newton Method has global convergence of O(1/k^(1+\nu)), but Theorem 4 shows the rate is O(1/k^(2+\nu)).

Correctness: correct

Clarity: well written

Relation to Prior Work: yes

Reproducibility: Yes

Additional Feedback: I decided to keep my score after rebuttals. I encourage the author conduct experiments about line-search methods in final version of this paper.

[Author Response · NeurIPS 2020]

We are very grateful to the reviewers for their invested time and expertise. Thank you very much for positive evaluation!
We hope that all raised issues are properly addressed in our rebuttal. All minor corrections are implemented.

**R1:** *Potential applications and significance.* In our opinion, one of the most notable applications of second-order
methods are ill-conditioned problems, which appear very often in practice. In particular, we are interested in mini-
mization of SoftMax objective (the log-sum-exp function). This is a smooth approximation of the pointwise maximum
of linear functions, and it is hard to solve this problem, when a smoothing parameter is small. At the same time, the
classical Newton method works badly on this problem in practice (having no global convergence guarantees), when the
starting point is far away from the optimum. We are happy to add our experimental results with this objective into the
supplementary part. Additionally, the current experiments with Logistic regression problem show that our stochastic
methods can be more efficient not only for reaching high-accuracy, but for low accuracy level ($10^{-2} - 10^{-4}$) as well
(see Fig. 6). We also believe, that our theoretical developments can help in constructing new second-order algorithms.

*Subsolver and Initialization.* We have a brief description of our subsolver in the suppl. part (lines 380-385). We will add
more details there. The code with our implementation will be available. The starting point was the origin. The first step
of FW algorithm (with the standard step-size $\gamma_k := \frac{2}{k+2}$) is on the border of the domain. We think, this is the reason of
the function value increase. The use of line-search in FW seems to be reasonable in this case. We will try that, thanks.

**R2:** *Contracting-Domain Newton vs. Aggregating Newton.* We agree that the first algorithm seems to have better
performance in practice. From our experience, the aggregating method is more stable though. We have a brief
comparison of these two methods in the supplementary section E (lines 373-379, and Fig. 5).

*Additional feedback.* Strong convexity of the composite term is assumed only in Theorem 2. One of the main
contributions of [29] was the extension of first-order conditional gradient method onto the case of composite optimization
problems. Additionally, the second-order Contracting Trust-Region method was proposed, which has the form of
Algorithm 2 from our work (however, only $O(1/k)$ rate was established). We will highlight these contributions in our
paper, thanks.

**R3:** *Stochastic methods* are both based on Algorithm 1. The difference between Algorithm 1 and 2 is in how we treat
the composite part (see the remark on line 123). We will try to make our presentation of stochastic variants more clear.

*The advantage over first-order methods.* The complexity of one step for simple sets can be estimated as $O(n^3)$ + the
cost of computing the Hessian. For Logistic regression, the cost of computing the gradient is $O(mn)$, and the Hessian
is $O(mn^2)$, where $m$ is the dataset size. Hence, when $m \gg n$, our methods can benefit significantly. Comparing the
convergence rates, our complexity parameter $H_\nu$ depends only on the variation of the Hessian (in arbitrary norm). It can
be much smaller than the max. eigenvalue of the Hessian, which typically appears in the rates of first-order methods.

*Numerical simulations.* We will provide more details regarding the experiments and the average performance of the
methods, thanks. There was a typo in (23), this rate should be the same as that one in (13). Note, that on Fig.3 and Fig.6
we compare the number of Epochs (total data accesses), not the iterations.

**R4:** *Bounded domain.* We think, that the assumption on the boundness of the problem domain is not very strong.
From the practical perspective, it seems natural to consider a bound for the variables (otherwise, the problem may
appear to be non-feasible). The domain can be often introduced on a stage of creating a machine learning model (for
example: regularization by $\ell_p$-ball, as in (4); or the standard simplex, which is the domain of the probabilities for finite
distributions). Considering the class of constrained optimization problems with bounded domain, our method has the
same global convergence rate as that one of Cubic Newton, for convex objectives with Lipschitz continuous Hessian. At
the same time, the subproblem at each iteration is simpler (no cubic terms). Thus, we think that our results are not weak.

*Better convergence rate.* Our methods can be considered as a second-order generalization of the conditional gradient
method. Up to our knowledge, the best convergence rates for the methods of this type are known to be $O(1/k)$, for the
class of convex functions with Lipschitz continuous derivatives (for example, [29]). The rate $O(1/k^2)$, which we prove
in our work, seems to be significantly better.

**R5:** *Hölder continuity.* The claims of our paper are valid for any $\nu \in [0, 1]$. It is important, that our methods do not use
a specific value of this parameter. Therefore, they may automatically adapt to the level of smoothness, achieving the
best complexity guarantee among all $\nu \in [0, 1]$. *The proof of (6)* is based on the Newton-Leibniz formula:

$$\|\nabla f(y) - \nabla f(x) - \nabla^2 f(x)(y-x)\|_* \;=\; \|\int_0^1 (\nabla^2 f(x + \tau(y-x)) - \nabla^2 f(x))(y-x)d\tau\|_* \;\overset{(5)}{\leq}\; \tfrac{H_\nu \|y-x\|^{1+\nu}}{1+\nu}.$$

*Missed references to Katyusha and SARAH:* thanks, we add them.

**R6:** *Empirical comparison with second-order methods with line search.* Indeed, it seems to be a reasonable and
interesting comparison. We are happy to add more experiments into the supplementary part. *A typo in the statement of
Theorem 4:* the correct rate should be $O(1/k^{1+\nu})$. This is fixed, thanks.

[Meta-Review · NeurIPS 2020]

Dear authors, Thank you for submitting your clear and well-written paper. I am pleased to report that all reviewers liked your paper and see a potential for the field. When producing the final camera-ready paper, please check the reviewer's remarks to make it stronger. Thank you